# Phage-encoded TelN inhibits bacterial Mre11-Rad50 nuclease to protect hairpin telomeres

Maya Houmel ⬤, Nicolas Pellaton, Anna Anchimiuk & Stephan Gruber ⬤ ✉

## Abstract

**Ends of linear chromosomes require protection from host repair machinery that otherwise will mistake them for damaged DNA. The *E. coli* bacteriophage N15 harbors a linear genome with covalently closed hairpin ends formed by the phage-encoded telomere resolvase TelN. The double-strand break repair complex Mre11-Rad50 (MR, SbcCD in *E. coli*) specifically targets DNA termini, yet how hairpin telomeres evade host nuclease degradation in bacteria remains unknown. Here, we demonstrate that TelN is essential and sufficient to protect N15 phage-derived hairpin telomeres from MR processing in *E. coli*. Using a combination of genetic and biochemical approaches, we show that this protective function requires both TelN sequence-specific DNA binding and species-specific protein-protein interactions. Notably, we found that protection is independent of TelN's resolution activity and does not require the C-terminal domains of TelN. Our findings reveal a potentially broad mechanism of telomere protection, providing insights into a conserved regulation of MR activity at chromosome ends across the tree of life.**

**Keywords** Telomere Protection; Mre11-Rad50 Nuclease; SbcCD; Telomere Resolvase; Linear Chromosomes
**Subject Categories** DNA Replication, Recombination & Repair; Microbiology, Virology & Host Pathogen Interaction

## Introduction

Genome integrity is essential for survival and propagation. For organisms with linear chromosomes, maintaining genomic termini presents two fundamental challenges across all domains of life. First, semi-conservative replication fails to fully duplicate chromosome termini, known as the end replication problem, resulting in the need for dedicated enzymes such as telomerase to prevent progressive sequence loss. Second, the ends of linear chromosomes must be protected from inappropriate processing by DNA repair machinery, referred to as the end protection problem. To address these fundamental challenges, eukaryotic cells have evolved specialized nucleoprotein structures called telomeres, to prevent failures in end replication and protection leading to nucleolytic

degradation, chromosome end-to-end fusion, and ultimately cell cycle arrest or cell death (De Lange, 2009; Lazzerini-Denchi and Sfeir, 2016).

While most bacterial and archaeal chromosomes are circular, lacking DNA ends, some prokaryotic organisms harbor linear replicons, requiring protective mechanisms to preserve genome integrity. Bacterial telomeres fall into two distinct categories (Chaconas and Kobryn, 2010; Volff and Altenbuchner, 2000). The first type comprises terminal inverted repeat (TIR) sequences bound by terminal proteins, as found, for instance, in *Streptomyces* (Bao and Cohen, 2001) and in the *Bacillus subtilis* phage phi29 (Yoshikawa and Ito, 1981). The other category consists of covalently closed hairpin DNA ends, identified in diverse organisms, including the Lyme spirochete *Borrelia burgdorferi* (Fraser et al, 1997), the plant pathogen *Agrobacterium tumefaciens* (Goodner et al, 2001) as well as in various bacteriophages such as *Klebsiella oxytoca* phage phiKO2 (Stoppel et al, 1995) and *E. coli* phage N15 (Rybchin and Svarchevsky, 1999).

The mechanism of hairpin telomere replication has been extensively studied in the *E. coli* bacteriophage N15. The N15 prophage does not integrate into the host bacterial genome but exists as a linear DNA molecule with covalently closed hairpin ends. During N15 replication, the telomere resolvase TelN linearizes a dimeric circular replication intermediate by cleaving at two specific telomeric junction sites and sealing the ends, generating two monomeric linear copies of the chromosome with covalently closed hairpin telomeres (Deneke et al, 2000; Ravin, 2003; Ravin et al, 2001). TelN cleaving-joining activity requires its cognate *ntelRL* sequence, a 56-bp palindromic core essential for TelN binding and catalytic activity with additional repeats found in the flanking DNA sequences (Deneke et al, 2002). TelN generates a linear DNA molecule with covalently closed hairpin ends designated *ntelL* (left end) and *ntelR* (right end), obtained from processing of *ntelRR* and *ntelLL* sites produced by DNA replication (Ravin, 2015). Notably, the N15 TelN-*ntelRL* system was used to linearize the 4.6 Mb *E. coli* circular genome, resulting in viable cells with either a single linear chromosome (Cui et al, 2007) or with two complementary linear fragments (Liang et al, 2013), harboring covalently closed hairpin telomeres. These cells exhibited normal physiology, aside from the dispensability of the chromosome dimer resolution pathway (*dif*, XerCD), as expected for linear chromosomes.

While TelN effectively addresses the end replication problem through hairpin telomere creation and maintenance, how these linear structures evade processing by host nucleases is unknown. A

Department of Fundamental Microbiology (DMF), Faculty of Biology and Medicine (FBM), University of Lausanne, 1015 Lausanne, Switzerland. ✉E-mail: stephan.gruber@unil.ch

significant threat to hairpin telomeres is the conserved Mre11-Rad50 (MR) complex, a key component of cellular double-strand break DNA repair machinery (Rojowska et al, 2014; Hopfner, 2023). The MR complex (also known as SbcCD in bacteria, including *E. coli*) comprises a Rad50 ATPase dimer associated with an Mre11 nuclease dimer, with an additional regulatory factor in eukaryotes (Nbs1 in mammals or Xrs2 in yeast) forming the MRN/MRX complex. MR specifically discriminates and processes linear DNA through a conserved mechanism in which Rad50 senses DNA ends, triggering ATP-dependent conformational changes that activate Mre11 nuclease activity (Käshammer et al, 2019); (Fig. EV1A). This enables both strand-specific endonucleolytic and 3′-5′ exonucleolytic cleavage at diverse DNA end structures, including free termini, protein-blocked ends, and hairpins (Connelly et al, 1998, 1999, 2003; Eykelenboom et al, 2008; Gut et al, 2022; Paull, 2018; Saathoff et al, 2018).

The MR complex, though essential for DNA repair, must be tightly regulated at telomeres. In eukaryotes, multiple parallel mechanisms protect chromosomal termini. Mammalian telomeres form compact T-loop structures through the shelterin complex, sequestering DNA ends (Van Ly et al, 2018; Smith et al, 2020), while in budding yeast, multiple Rap1 proteins sterically bind and cap telomeric regions, presumably stiffening DNA (Le Bihan et al, 2013). Simultaneously, three evolutionarily distinct peptide motifs are known to inhibit MRN/MRX directly. The iDDR motif in mammalian shelterin protein TRF2, MIN in telomere-associated Taz1 (fission yeast), and BAT in telomere-associated Rif2 (budding yeast), all target the same β-sheet of RAD50 to block nucleolytic processing at chromosome ends (Bombarde et al, 2010; Fan et al, 2025; Khayat et al, 2021, 2024; Marsella et al, 2021; Myler et al, 2023; Okamoto et al, 2013; Roisné-Hamelin et al, 2021).

Similar protection strategies against MR are needed in prokaryotes with linear replicons but remain largely uncharacterized. Hairpin DNA is efficiently processed by the MR complex (Saathoff et al, 2018); thus, a mechanism for hairpin telomere protection from inadvertent degradation must be in place for N15 propagation. In this study, we discover a novel role for TelN in hairpin telomere protection from *E. coli* MR processing, which depends on both sequence-specific DNA binding and species-specific protein–protein interactions. This protective function is distinct from TelN's previously known catalytic activity and independent of its C-terminal domains. Our data suggest that MR inhibition at telomeres may represent one of the earliest evolutionary solutions to the challenge of maintaining linear chromosomes, establishing fundamental principles of telomere protection conserved from bacteria to humans.

# Results

## Chromosome linearization is lethal in the presence of host DNA repair nuclease Mre11-Rad50 in *B. subtilis*

To assess whether bacterial chromosomes can be linearized across different species, we attempted to generate linear chromosomes in *B. subtilis* using the phage-encoded telomere resolvase TelN and its *ntelRL* core sequence extended by neighboring sequences from the N15 phage (Fig. 1A), an approach previously shown to successfully linearize the *E. coli* chromosome (Cui et al, 2007). We therefore engineered a *B. subtilis* strain containing this *ntelRL* sequence, denoted as WT*ntelRL*. Despite multiple attempts, no clones carrying both *telN* and intact *ntelRL* could be obtained (Figs. 1B and EV1B), with rare suppressor clones carrying various deletions or mutations in the *ntelRL* insertion site (Fig. EV1C). We wondered whether the generated linear chromosomes might be unstable due to their targeting by host DNA repair nucleases. To test this, we repeated the experiment in a strain lacking a gene for Mre11-Rad50 (ΔMR*ntelRL*). Strikingly, transformation with a xylose-inducible *telN* construct yielded significantly more colonies in the ΔMR*ntelRL* strain compared to WT*ntelRL*, where only a few, small-sized colonies were observed (Figs. 1B and EV1B). These results suggest that MR actively prevents the formation or maintenance of linear chromosomes in *B. subtilis*, but not in *E. coli* (Cui et al, 2007).

To confirm and further characterize linear chromosome formation in *B. subtilis*, we first used antibiotic sensitivity as a proxy for genomic linearization, employing a spectinomycin resistance gene inserted at *ntelRL* which becomes sensitized to the DNA hairpin formation likely due to loss of DNA super helicity near the end (Fig. 1A). We observed two distinctive growth phenotypes in *B. subtilis* strains with linearized chromosomes. First, strains with a linear chromosome displayed impaired growth and the frequent appearance of suppressors in the absence of xylose, likely due to a requirement of TelN for telomere resolution (Fig. 1C). Second, these strains showed reduced spectinomycin resistance, consistent with lowered gene expression at the DNA end, likely due to loss of DNA underwinding (Fig. 1C), as demonstrated for linear versus circular plasmids in *Borrelia burgdorferi* where topological changes directly affect transcriptional activity (Beaurepaire and Chaconas, 2007). Moreover, we performed Southern blotting to confirm the linearization of *ntelRL* DNA on the *B. subtilis* chromosome (Fig. EV1D,E). Taken together, these data provide strong evidence for successful chromosome linearization in *B. subtilis* strains lacking Mre11-Rad50 but not in the wild type.

## TelN is required for the protection of hairpin telomeres in *E. coli*

These results identify MR as a fundamental barrier to linear chromosome formation or maintenance in *B. subtilis*. However, the successful linearization of *E. coli* chromosomes using the same TelN-*ntelRL* system (Cui et al, 2007) suggests that in *E. coli*, this barrier does not exist or that it is overcome by dedicated mechanisms. We therefore next investigated how *E. coli* stably maintains linear DNA molecules despite the presence of MR. As a simple model system, we chose the phage N15-derived linear plasmid (previously commercialized under the name pJAZZ), here denoted as pLIN_*ntelN* (Godiska et al, 2009). pLIN_*ntelN* carries its own copy of the *telN* gene necessary for its replication as well as a chloramphenicol-resistant marker for selection of transformants (Fig. EV1F). We first assessed the transformation efficiency of *E. coli* host cells containing or lacking an additional *telN* gene. Briefly, *telN* was placed under an arabinose-inducible promoter ($P_{BAD}$) and integrated into the *E. coli* K12 chromosome near the *glmS* locus by mini-Tn7 transposon-based insertion (Bao et al, 1991). Strikingly, in the absence of chromosome-encoded *telN*, transformation of linear DNA was severely impaired compared to a circular control plasmid (Fig. 1D), consistent with previous observations that linear

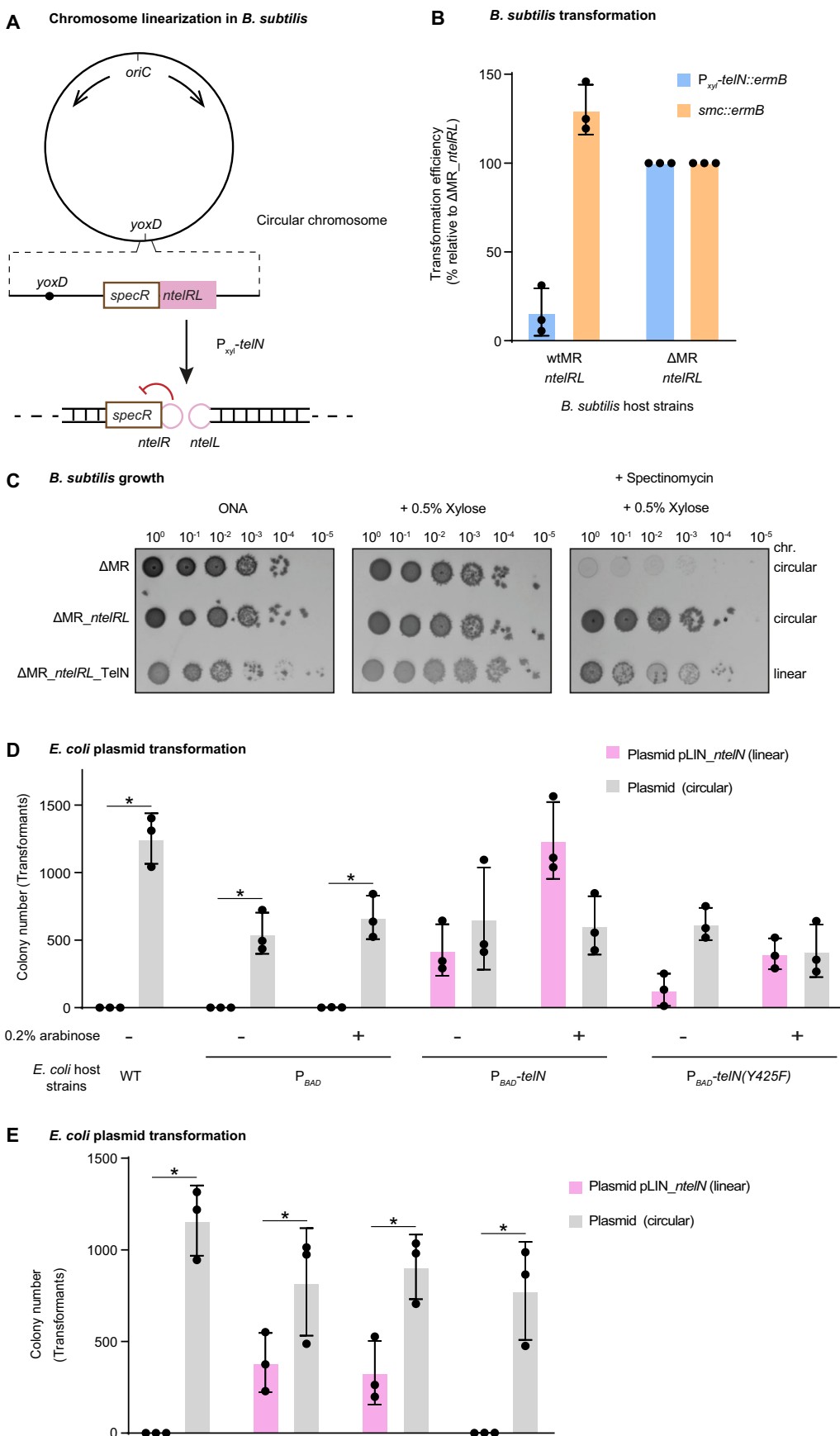

◄ **Figure 1.   Antagonistic roles of Mre11-Rad50 and TelN in maintenance of linear DNA with hairpin ends.**

(A) Schematic depiction of chromosome linearization in *B. subtilis*. Upon induction of TelN expression with xylose (P*xyl*-telN), TelN recombines *ntelRL* inserted at *yoxD* using its cleaving-joining activity, resulting in a linear *B. subtilis* chromosome with hairpin termini. The formation of these hairpin telomeres locally alters gene expression, resulting in reduced spectinomycin resistance (red arrow). (B) Counts of erythromycin-resistant colonies for the indicated *B. subtilis* backgrounds (WT*ntelRL* and ΔMR*ntelRL*) after transformation with *telN* (P*xyl*-telN::*ermB*) or *smc* (*smc::ermB*) constructs. Data were normalized to the ΔMR*ntelRL* strain (set as 100%). Means and standard deviations obtained from three biological replicates are shown. (C) Serial spot dilutions (10⁰–10⁻⁵) of *B. subtilis* ΔMR strains carrying either a circular or linear chromosome: ΔMR, ΔMR*ntelRL*, and ΔMR*ntelRL*_TelN. Cells were spotted on nutrient agar (ONA) plates supplemented as indicated. Growth was assessed under three conditions: ONA, ONA + 0.5% xylose, and ONA + spectinomycin 50 μg/mL + 0.5% xylose. (D) Expression of TelN promotes transformation with linear pLIN plasmid. Counts of chloramphenicol-resistant colonies for the indicated *E. coli* backgrounds after transformation with either a linear plasmid with *ntel* hairpin ends (*ntelL* and *ntelR*), pLIN_*ntelN*, or a control circular plasmid. Strains were grown with (+) or without (−) 0.2% arabinose. WT: wild-type strain, P*BAD*: WT strain with chromosomal integration of arabinose-inducible promoter alone, P*BAD*-telN: WT strain with chromosomal integration of arabinose-inducible TelN, P*BAD*-telN(Y425F): WT strain with chromosomal integration of arabinose-inducible TelN(Y425F). Each transformation from three biological replicates was plated separately, and colonies were counted as the experimental unit. Sample size was chosen based on prior experience with similar assays, where three independent replicates consistently captured the variability in transformation efficiency. Colony counts from replicate plates were used to calculate the mean and standard deviation. Asterisks indicate samples (circular vs linear) with a *p* value, obtained by paired *t*-tests, lower than 0.05. WT: *p* = 0.0073; P*BAD* without arabinose: *p* = 0.0245; P*BAD* with arabinose: *p* = 0.019. Note that data for selected samples are also shown in Fig. 3B. (E) Counts of chloramphenicol-resistant colonies for the indicated *E. coli* backgrounds after transformation with either pLIN_*ntelN* or a control circular plasmid. Means and standard deviations from three biological replicates are shown. Asterisks indicate samples (circular vs linear) with a *p* value, obtained by paired *t*-tests, lower than 0.05. WT: *p* = 0.009; Δ*rad50*: *p* = 0.0471; Δ*mre11*: *p* = 0.0144; Δ*mutS*: *p* = 0.0372.

DNA with hairpin telomeres transforms *E. coli* with ~100-fold lower efficiency than circular DNA (Dorokhov et al, 2004). However, transformation of linear and circular DNA was similarly efficient when TelN was already present in host cells prior to transformation (Fig. 1D), consistent with the previous observation that the presence of the N15 prophage or of TelN-expressing plasmids increases transformation efficiency with linear DNA (Dorokhov et al, 2004). The presence of host-encoded *telN* is therefore needed for efficient uptake or maintenance of the linear plasmid with hairpin telomeres, presumably until plasmid-encoded TelN accumulates to sufficient levels.

Based on our previous findings in *B. subtilis* (Fig. 1B), we suspected that the *E. coli* MR may be responsible for the low transformation efficiency of pLIN_*ntelN* in the absence of host-encoded *telN*. Thus, we transformed both linear and circular plasmids into *E. coli* strains deficient in the *rad50* or *mre11* genes. We observed that the absence of the *rad50* or *mre11* genes resulted in increased transformation efficiency of pLIN_*ntelN*, almost to the level obtained by transformation with a circular plasmid, even in the absence of host-encoded *telN* (Fig. 1E). This is not the case when deleting an unrelated gene, such as *mutS*, in the host strain (Fig. 1E). These results demonstrate that MR is indeed responsible for the low transformation efficiency of pLIN_*ntelN* and suggest that TelN protects linear DNA from MR-mediated degradation in *E. coli*.

### Inhibition of Mre11-Rad50 by TelN in vitro

To determine whether TelN inhibits Mre11-Rad50 nuclease activity in the absence of other factors, we developed an in vitro DNA degradation assay with a fluorescently labeled linear DNA substrate with a hairpin end. This substrate was generated by PCR amplification of the *ntelRL* sequence, along with adjacent N15 phage genomic regions, using a 5′ biotinylated primer labeled with fluorescein (Fig. 2A). TelN-mediated resolution of the *ntelRL* PCR product yielded fragments with one hairpin end. The fluorescently labeled hairpin substrate was immobilized on streptavidin-coated magnetic beads, using the physical size of the beads to sterically protect the biotin-labeled non-hairpin end and thus allowing MR to

process the DNA from only one side, mimicking its suspected activity at unprotected telomeres (Fig. 2A).

We added purified MR and TelN protein (Fig. EV2A,B) to the bead-immobilized hairpin substrate and observed that in the absence of TelN, MR efficiently degraded the DNA in the presence of ATP (Figs. 2B and EV3A,B). Interestingly, the addition of TelN inhibited MR nuclease activity in a dose-dependent manner (Figs. 2B and EV3A,B). This demonstrates that TelN is sufficient to protect hairpin telomeres from *E. coli* MR degradation.

### Hairpin telomere protection is independent of TelN's ability to resolve hairpin telomeres

Given that TelN is primarily characterized as a telomere resolvase, we next investigated whether its catalytic activity was required for hairpin telomere protection. We thus generated a catalytically inactive TelN(Y425F) mutant using site-directed mutagenesis (Deneke et al, 2000) (Fig. EV2B). As expected, purified TelN(Y425F) showed no detectable DNA resolution activity on the *ntelRL* target sequence even at high protein concentrations, while TelN WT efficiently resolved the substrate (Fig. EV2C,D). Strikingly, we observed that TelN(Y425F) inhibited MR-mediated hairpin degradation as effectively as TelN WT in our in vitro protection assay (Figs. 2C and EV3C). This suggests that TelN's resolvase and protection activities can be separated. To confirm these observations in a cellular context, we expressed TelN(Y425F) from the *E. coli* chromosome and assessed its impact on telomere protection. Interestingly, TelN(Y425F) improved the transformation efficiency of linear plasmid with hairpin ends similar to WT (Fig. 1D). This result further supports the notion that TelN's protective function is mechanistically distinct from its telomere resolution activity.

### DNA end binding by TelN

To understand how TelN mediates protection, we first assessed its DNA binding properties by fluorescence anisotropy. We designed fluorescein-labeled DNA substrates containing the *ntelR* sequence in two conformations: a 40-bp *ntelR* DNA hairpin and an 80-bp

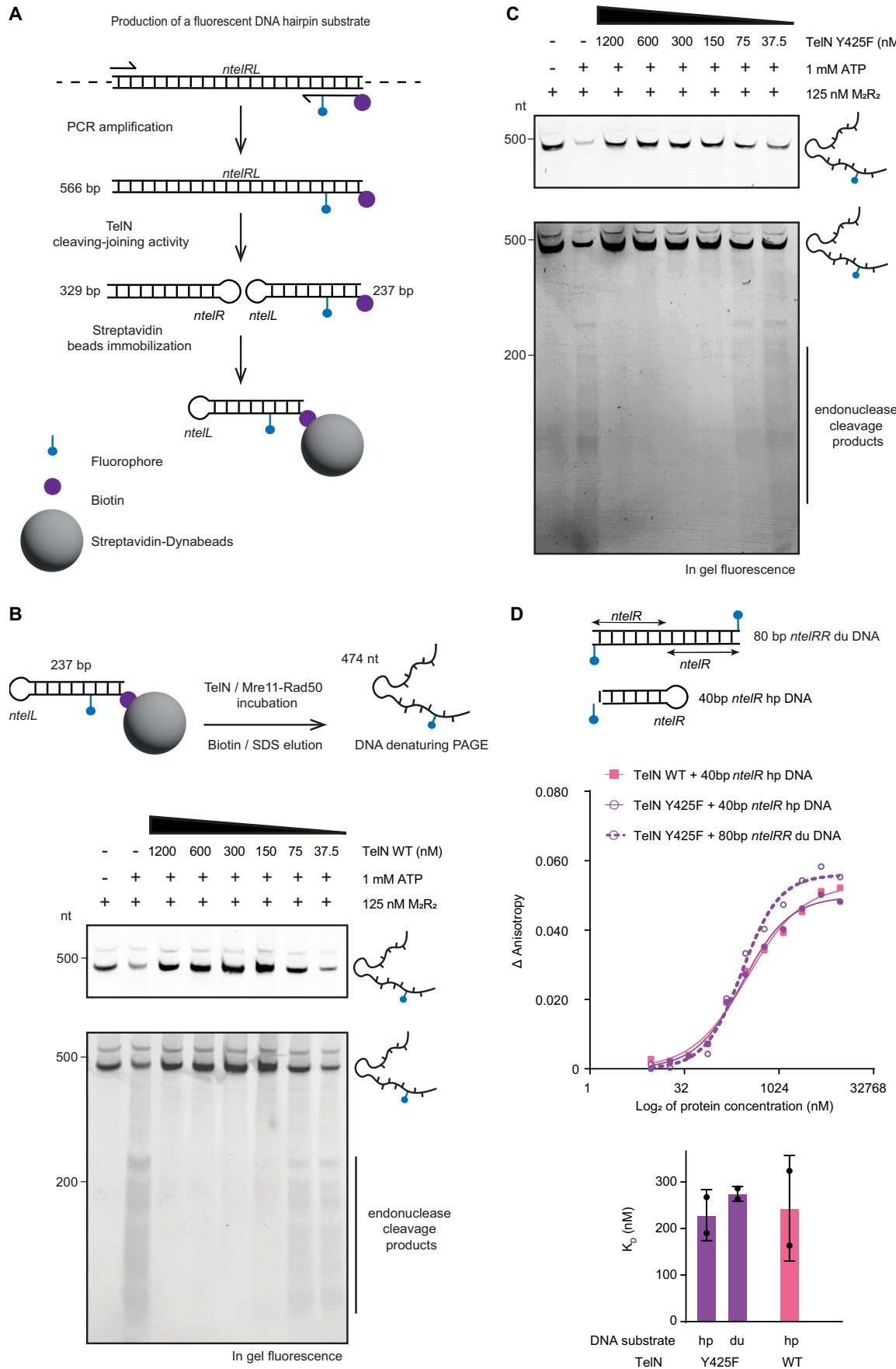

**Figure 2.  TelN directly inhibits Mre11-Rad50 nuclease activity independently of its catalytic function.**

(A) Schematic representation of the generation of a fluorescent hairpin DNA substrate for in vitro protection assays. (B, C) In vitro protection assay with *E. coli* Mre11-Rad50 using (B) TelN WT or (C) TelN(Y425F). DNA degradation products were analyzed by DNA-denaturing PAGE after biotin/SDS elution. Lower panel: representative in-gel fluorescence analysis showing DNA degradation profile. Upper panel: same gel image with reduced exposure optimized to visualize remaining substrate bands. (D) DNA binding analysis by fluorescence anisotropy. Upper panel: binding curves of TelN WT with 40 bp *ntelR* DNA hairpin ("hp") and TelN(Y425F) with hp and with 80 bp *ntelRR* DNA duplex ("du"). Lower panel: calculated dissociation constants ($K_D$) for different DNA substrates and TelN variants. Data points and error bars represent means and standard deviations from two technical replicates ($n = 2$).

linear *ntelRR* DNA duplex (Fig. EV3D). TelN bound to these substrates with moderate affinity (dissociation constant ($K_D$) in the ~200–300 nM range), regardless of the DNA structure or length (Fig. 2D). Due to the catalytic activity of TelN, its binding to the linear 80 bp *ntelRR* duplex DNA substrate could be affected by the resolution of the DNA. As expected, no significant differences in binding affinity were observed between TelN WT and TelN(Y425F) on the 40 bp *ntelR* hairpin substrate, suggesting that the catalytic activity of TelN does not influence its DNA binding properties to a single binding site (Fig. 2D). Surprisingly, the binding affinity of TelN(Y425F) remained comparable between the hairpin with a single binding site and the DNA duplex with two binding sites, indicating a lack of strong cooperativity in DNA binding and implying that two TelN proteins bind largely individually to the two binding sites (Fig. 2D). We conclude that TelN recognizes its target sequence with moderate affinity. Our findings suggest that TelN binds to the educt and the product of the enzymatic reaction with similar affinity, implying that it remains bound to the target DNA after telomere resolution, unlike typical enzymes. This behavior is consistent with observations with related telomere resolvases: ResT from *B. burgdorferi* binds hairpin telomeres and can promote telomere fusion, the reverse reaction (Kobryn and Chaconas, 2005), structural studies reveal an intact dimer of TelA binding to hairpin products (Shi et al, 2013), and TelK shows limited turnover on hairpin DNA products (Huang et al, 2004).

## The C-terminal domains of TelN are dispensable for hairpin telomere protection

The C-terminal domains of telomere resolvases are less well conserved across species, and the functions remain unknown (Fig. EV4A). To investigate their role in TelN's protective function, we generated three truncation variants lacking the C-terminal domain(s): Δ445–631, Δ541–631, and Δ583–631 (Fig. 3A). We integrated these mutants into the *E. coli* chromosome under arabinose-inducible control to assess their ability to protect linear DNA in vivo. TelN WT provided protection even without induction. All truncation variants, including the deletion of both C-terminal domains (Δ445–631), protected linear DNA with hairpin ends but required arabinose induction to restore the transformation efficiency of linear plasmids to (near) wild-type levels (Figs. 3B and EV4B).

We also tested whether some of these truncated variants could protect hairpin telomeres from MR degradation in vitro. Purified Δ541–631 and Δ583–631 proteins (Fig. EV4C) inhibited MR nuclease activity comparably to wild-type TelN (Figs. 3C,D and EV5A,B), while retaining full catalytic activity in telomere resolution assays (Fig. EV4D,E). These results demonstrate that the C-terminal sequences are dispensable for telomere

protection, suggesting that the core protection mechanism resides in the more conserved region of the protein.

## Sequence specificity of TelN-mediated protection is modulated by cellular context

To elucidate the molecular requirements for TelN-mediated protection, we tested whether binding of TelN to *ntelRL* was important for protection. We first tested in vivo whether TelN could protect non-cognate DNA sequences by comparing it with TelPY, a related telomere resolvase from *Yersinia enterocolitica* bacteriophage PY54 (Hertwig et al, 2003) (Fig. EV4A). We designed linear plasmids with two different hairpin ends for analysis (Fig. 4A). Starting from pLIN_*ntelN*, we produced a circular form upon inactivation of the plasmid-encoded *telN* gene through an early stop codon and in the same cloning step, ligating *ntelR* and *ntelL* to recreate *ntelRL*. We then replaced the *ntelRL* recognition sequence by *pytelRR*, the recognition sequence of TelPY, which shares 59.5% sequence identity with *ntelRL* (Fig. EV6A). This plasmid was transformed into a ΔMR *E. coli* strain expressing chromosomal *telPY*, yielding a linear plasmid with two *pytelR* hairpin ends (Fig. 4A).

Using the two linear constructs, pLIN_*ntelN* with *ntel* and pLIN_*pytel* with *pytel* hairpin ends, we assessed protection specificity through plasmid transformation assays. Despite the sequence similarity between their respective binding sites, the proteins exclusively protected their cognate DNA sequence. Chromosomally encoded *telN* enabled efficient transformation of pLIN_*ntelN* even without arabinose induction, but failed to support transformation of pLIN_*pytel* (Fig. 4B) even with induction. Similarly, TelPY showed similar sequence specificity by only protecting plasmids with its cognate *pytel* hairpin ends, though requiring arabinose induction for protection (Fig. 4B). This strict specificity suggests that sequence-specific DNA binding is essential for protection, ruling out mechanisms that solely rely on a direct interaction between the telomere resolvase protein and the MR complex. Notably, TelPY protects (*pytel*) hairpin ends from *E. coli* MR despite its higher sequence divergence from TelN (Fig. EV4A), possibly by steric hindrance rather than specific protein–protein contacts.

To gain more insights into the DNA specificity, we examined the protective capacity of TelN under defined biochemical conditions. We assessed TelN's ability to protect three distinct DNA substrates in vitro: two fluorescently labeled *ntel* hairpins (*ntelL* and *ntelR*) and a nonspecific fluorescent linear DNA fragment lacking *ntelRL* sequences (Fig. 4C). Interestingly, at slightly elevated protein concentrations, TelN also protected nonspecific DNA ends from MR degradation, with nearly complete protection of all substrates at 640 nM TelN (Figs. 4C and EV6B–D). This discrepancy between

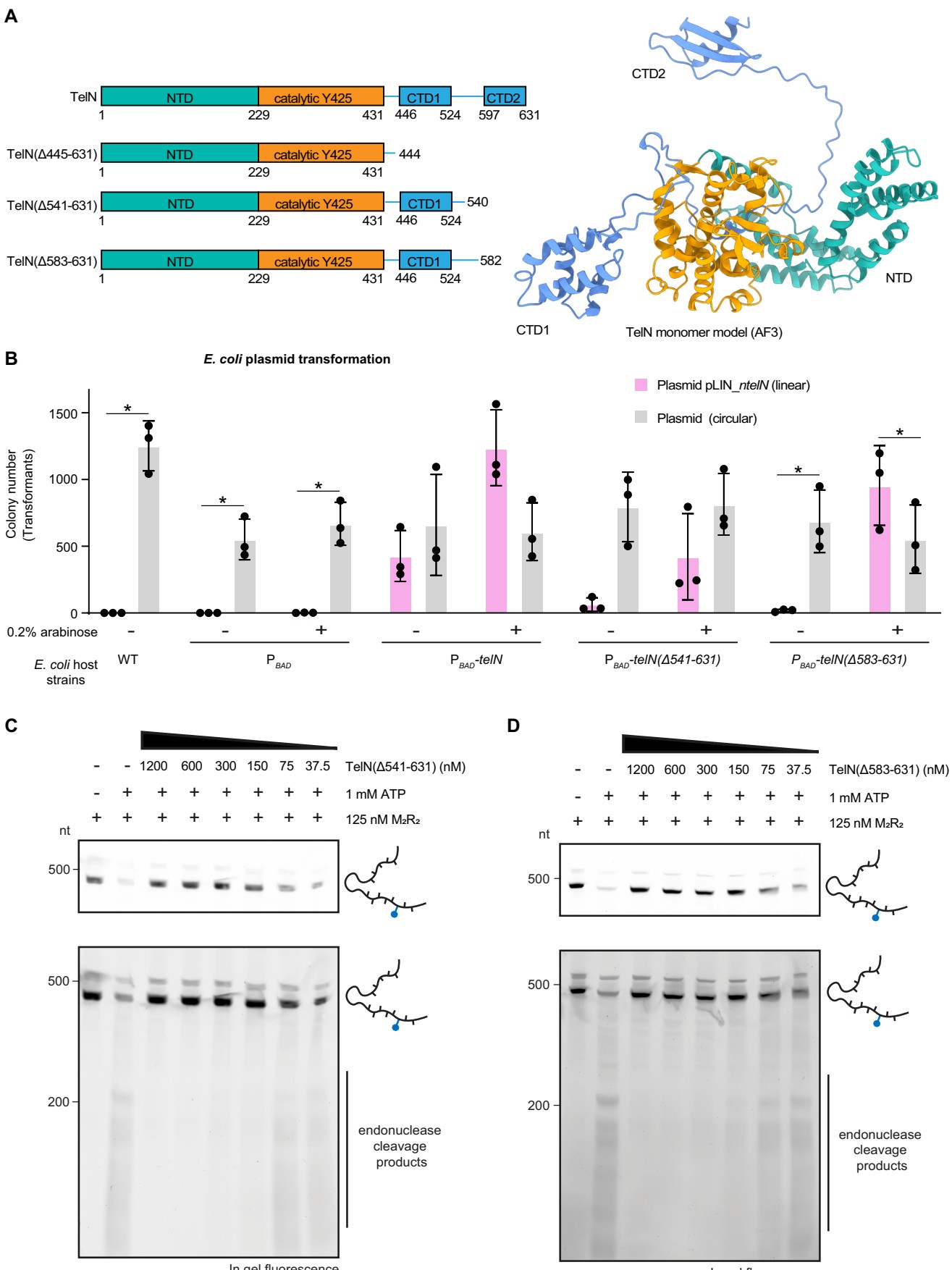

**A**

**B** *E. coli* plasmid transformation

Plasmid pLIN_*ntelN* (linear)

Plasmid (circular)

0.2% arabinose

*E. coli* host strains: WT, P<sub>BAD</sub>, P<sub>BAD</sub>-*telN*, P<sub>BAD</sub>-*telN(Δ541-631)*, P<sub>BAD</sub>-*telN(Δ583-631)*

**C** In gel fluorescence

**D** In gel fluorescence

**Figure 3. The C-terminal domains of TelN are dispensable for telomere protection.**

(A) Left panel: Domain organization of TelN with positions of truncations (Δ445–631, Δ541–631, and Δ583–631). Right panel: Alphafold3 prediction of a TelN monomer (pTM = 0.74) showing N-terminal domain in green (NTD), catalytic domain in orange, and C-terminal domains in blue (CTD1 and CTD2). (B) Counts of chloramphenicol-resistant colonies for the indicated *E. coli* strains after transformation with pLIN_*ntelN* or a control circular plasmid. Strains were grown with (+) or without (−) 0.2% arabinose. $P_{BAD}$-*telN(Δ541–631)*: WT strain with chromosomal integration of TelN(Δ541–631), $P_{BAD}$-*telNΔ583–631*: WT strain with chromosomal integration of arabinose-inducible TelN(Δ583–631). Means and standard deviations from three biological replicates are shown. Asterisks indicate samples (circular vs linear) with a *p* value, obtained by paired *t*-tests, lower than 0.05. WT: $p = 0.0073$; $P_{BAD}$ without arabinose: $p = 0.0245$; $P_{BAD}$ with arabinose: $p = 0.019$; $P_{BAD}$-*telN(Δ583–631)* without arabinose: $p = 0.0370$; $P_{BAD}$-*telN(Δ583–631)* with arabinose: $p = 0.0305$. Same experiment as in Fig. 1D, displaying additional data for *telN* truncation mutants and controls. A similar experiment using TelN(Δ445–631) is shown in Fig. EV4B. (C, D) DNA protection using purified TelN(Δ541–631) protein (C) and TelN(Δ583–631) protein (D). DNA degradation products were analyzed by DNA-denaturing PAGE. Lower panel: representative in-gel fluorescence analysis showing DNA degradation profile. Upper panel: same gel image with reduced exposure optimized to visualize remaining substrate bands.

in vivo specificity and broader substrate protection in vitro suggests that spatial organization likely modulates TelN's protective function in cells. While TelN can inhibit MR activity on various DNA substrates when present at high concentrations in solution, the cellular environment, with its vast excess of non-cognate DNA, likely constrains this activity through sequence-specific recruitment, ensuring protection is directed specifically to telomeric sequences, while DNA repair can occur elsewhere in the genome.

## Species specificity of TelN-mediated protection

To further characterize the protection mechanism, we investigated whether TelN could inhibit MR complexes from different species. We compared TelN's ability to inhibit nucleolytic processing by both *E. coli* MR and a eukaryotic counterpart, the MRX complex from *S. cerevisiae*. Despite their functional similarity, these complexes only share limited sequence homology. *E. coli* MR proteins share only limited sequence similarity with their yeast counterparts. Using a fluorescently labeled hairpin substrate immobilized on streptavidin-coated magnetic beads, we compared TelN's protective activity against purified *E. coli* MR versus its eukaryotic ortholog MRX. TelN effectively blocked *E. coli* MR nuclease activity, while showing no protection against yeast MRX degradation (Fig. 5A,B). This species-specificity suggests that TelN makes specific contacts with the bacterial MR complex, pointing toward a protection mechanism involving direct protein–protein interactions with the *E. coli* complex, thus providing an explanation for the lack of protection observed in *B. subtilis*. Alternatively, or additionally, telomere resolvases may hinder bacterial MR by steric hindrance but not yeast MRX, for instance, due to the greater length of its coiled coils.

## Discussion

Maintenance of chromosome ends and protection from cellular nucleases pose a fundamental challenge for organisms with linear chromosomes. Eukaryotes have evolved several independent mechanisms to protect their telomeres from degradation, prevent end-to-end chromosome fusions, and telomere erosion. In contrast, the mechanisms governing DNA end protection in bacteria with linear chromosomes remain poorly understood. This study demonstrates the role of phage-encoded TelN telomere resolvase in hairpin telomere protection. Using both in vivo and in vitro approaches, we show that TelN is essential and sufficient to protect

hairpin telomeres from MR nuclease degradation in *E. coli* (Figs. 1D and 2B). Notably, the hairpin resolvase catalytic activity of host-encoded TelN is not needed for efficient protection of linear DNA with hairpin ends (Figs. 1D and 2C), revealing a clear separation of function.

## Molecular mechanisms underlying TelN-mediated end protection

Our findings suggest that TelN protects chromosome ends through multiple coordinated steps. The first step involves the formation of a telomeric hairpin by TelN itself, which, in addition to solving the DNA end replication problem, eliminates free 3' and 5' ends that would otherwise be susceptible to exonuclease activity, such as from RecBCD. Second, our data demonstrate that sequence-specific DNA recognition and binding are required, as TelN protects only its cognate sequence from MR in vivo (Fig. 4B). Third, the specific inhibition of *E. coli* MR, but not yeast MRX, points to a direct protein–protein interaction between TelN and the *E. coli* MR complex (Fig. 5), possibly in addition to steric blockage as indicated by TelPY's inhibition of *E. coli* MR. Together, these results suggest that at least two features, DNA binding and MR interaction, act in concert to inhibit MR. Loss of either component permits MR to cleave the hairpin, likely enabling further degradation by exonucleases such as RecBCD. This dual requirement mirrors strategies employed by eukaryotic telomere protection factors. For instance, in *S. cerevisiae*, the telomere-associated protein Rap1 recognizes, and binds repeat sequences in tandem, presumably stiffening telomeric DNA (Williams et al, 2010; Le Bihan et al, 2013). Rap1 also recruits another telomere-associated protein, Rif2, which directly inhibits MRX through its BAT motif binding to Rad50 (Roisné-Hamelin et al, 2021).

The observed difference between TelN's sequence-specific protection in vivo and its ability to protect nonspecific DNA in vitro (Fig. 4B,C) provides important insights into the spatial regulation of telomere protection. While TelN can inhibit MR activity on various DNA substrates at elevated concentrations in solution, its protective function requires precise binding to specific sequences in a cellular context. This mirrors the eukaryotic telomere protection factor Rif2, which inhibits MRX endonuclease activity in vitro without requiring DNA recruitment (Marsella et al, 2021), yet in vivo, it is spatially constrained through Rap1-dependent localization at telomeres (Roisné-Hamelin et al, 2021). This permits the distinction between chromosome ends that need to be protected from DNA damage and sites that need to be

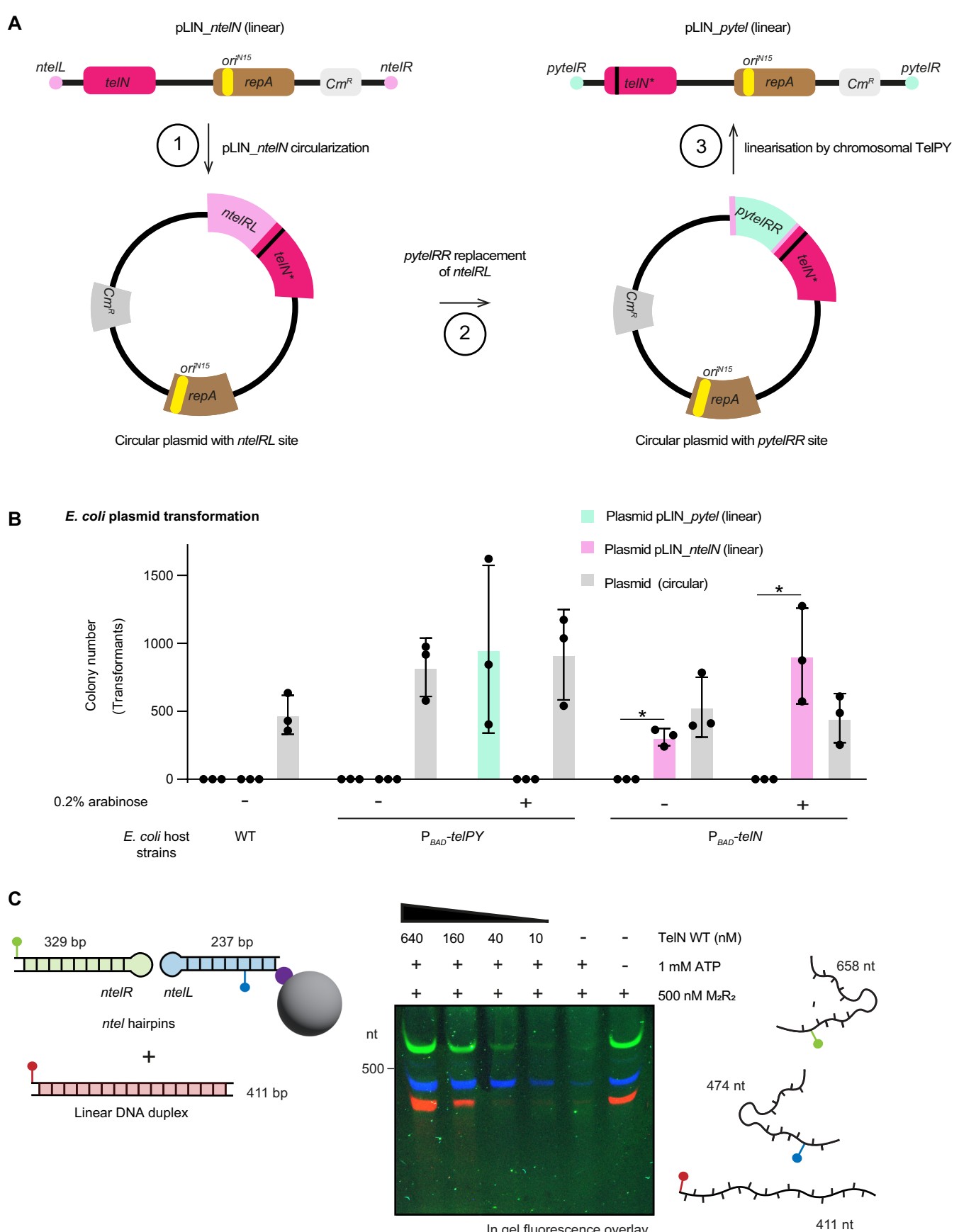

**A**

pLIN_*ntelN* (linear)

pLIN_*pytel* (linear)

**B**  ***E. coli* plasmid transformation**

- Plasmid pLIN_*pytel* (linear)
- Plasmid pLIN_*ntelN* (linear)
- Plasmid (circular)

**C**

*ntel* hairpins

Linear DNA duplex

In gel fluorescence overlay
DNA denaturing PAGE

**Figure 4. Sequence specificity of TelN-mediated DNA hairpin protection.**

(A) Overview of the cloning procedure for generating linear plasmids pLIN_pytel from pLIN_ntelN. Step 1: telN mutagenesis to generate a circular plasmid containing the unresolved ntelRL sequence. Step 2: Replacement of ntelRL with pytelRR sequence. Step 3: TelPY-mediated linearization to generate a linear plasmid pLIN_pytel. Adapted from (Liu et al, 2022). (B) Counts of chloramphenicol-resistant colonies for the indicated E. coli strains after transformation with linear plasmids pLIN_ntelN and pLIN_pytel or a control circular plasmid. Means and standard deviations from three biological replicates are shown. Asterisks indicate samples (linear DNAs) with a p value, obtained by paired t-tests, lower than 0.05. $P_{BAD}$-telN without arabinose: $p = 0.0136$; $P_{BAD}$-telN with arabinose: $p = 0.0469$. (C) Left panel: Schematic representation of DNA degradation substrates. Three distinct fluorescently labeled DNA substrates were tested simultaneously: two fluorescently labeled ntel hairpin substrates of 329 and 237 bp, and one nonspecific linear DNA fragment of 411 bp. Right panel: Representative example of an in-gel fluorescence analysis showing DNA products. Image overlay displaying three substrates in different colors (green/blue: ntel hairpin substrates appearing at 658 and 474 bp due to denaturation; red: nonspecific linear DNA at 411 bp).

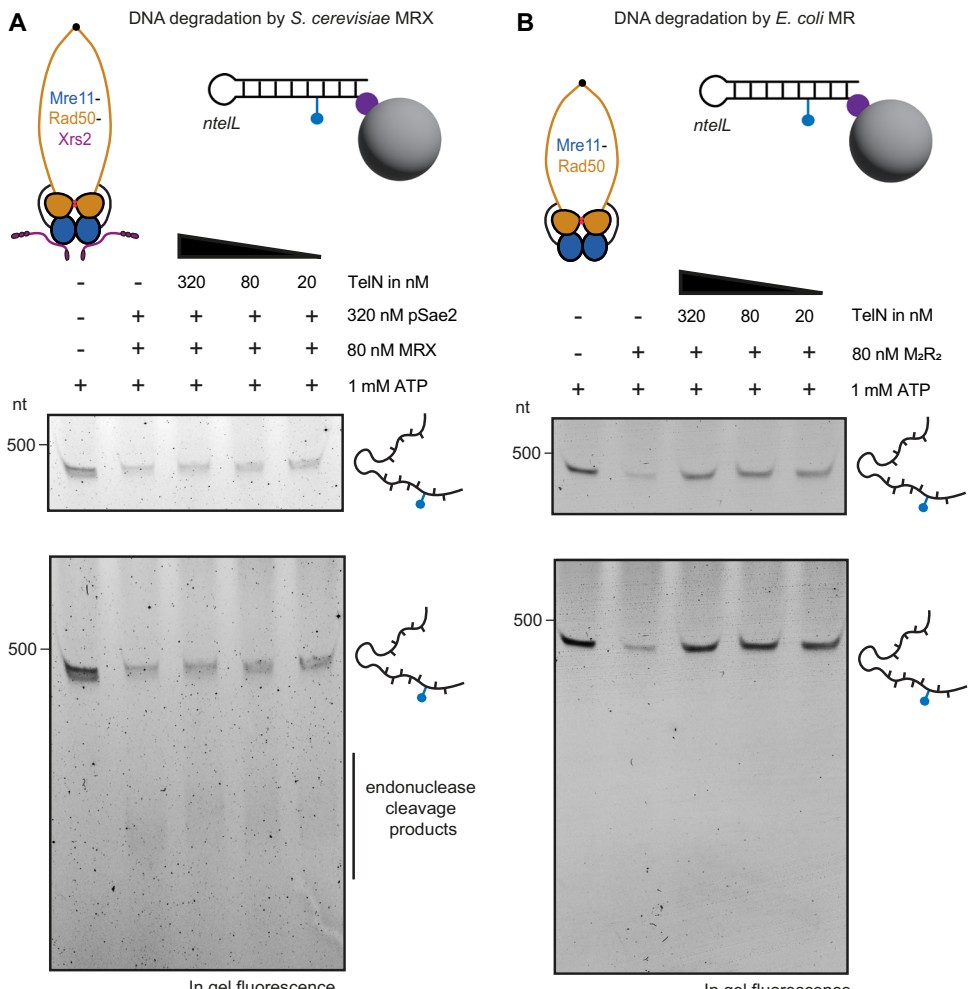

**Figure 5. TelN-mediated protection exhibits species specificity.**

(A) Hairpin DNA degradation by S. cerevisiae MRX complex. Lower panel: representative in-gel fluorescence analysis showing DNA degradation profile after incubation with pSae2 and MRX. Upper panel: same gel image with exposure optimized to visualize remaining substrate bands. (B) DNA protection by TelN from degradation of E. coli MR complex. Lower panel: representative in-gel fluorescence analysis showing DNA degradation profile after incubation with MR. Upper panel: same gel image with exposure optimized to visualize remaining substrate bands.

repaired. Likely, TelN remains bound to newly generated DNA ends, thus ensuring protection of DNA ends even at low TelN concentrations.

Recent structural insights into MR conformational states during DNA end processing show that recognition of a DNA end induces a ring-to-rod transition of the coiled coils following ATP hydrolysis (Käshammer et al, 2019; Gut et al, 2022). The Mre11 dimer moves to the side, forming a nuclease proficient complex. TelN might hamper the conformational change of the MR complex, preventing ATP binding and hydrolysis, thus blocking the release of the nuclease and preventing DNA end processing. MR would be stuck in its ATP-bound state, unable to capture telomeric structures. Our in vitro assays support this hypothesis, demonstrating that TelN directly inhibits the nuclease activity of the MR complex (Fig. 2B),

thereby protecting DNA ends from degradation. A similar mechanism is proposed in eukaryotic systems, where, for instance, the iDDR motif in mammalian shelterin protein TRF2, MIN in telomere-associated Taz1 (fission yeast), and BAT in Rif2 (budding yeast) inhibit MRN/X-dependent DNA resection at telomeres by binding the same exposed β-sheet region of the RAD50 ATPase head (Fan et al, 2025; Khayat et al, 2024). The structural basis of potential TelN-MR interactions remains to be determined, and high-resolution structural studies using cryo-electron microscopy could reveal whether TelN induces conformational changes in the *E. coli* MR complex. Our attempts to detect direct interactions through co-immunoprecipitation between TelN and MR were unsuccessful, suggesting that their interaction might be transient or context-dependent.

## Higher-order structure formation in DNA end protection

Several observations suggest that TelN may induce structural changes in telomeric DNA for efficient protection. The natural target sequence of TelN appears significantly larger (310 bp) than the recognition sites required for efficient recombination (*ntelRL*, *ntelRR*, or *ntelLL*, 56 bp) (Deneke et al, 2002). These recognition sites contain the core *telO* sequence (28 bp) plus L3 and R3 sequence motifs (each 10 bp) flanking the core (Deneke et al, 2000). At least two degenerate copies of the L3/R3 motifs (L1 and L2 or R1 and R2, respectively) are found in the neighboring regions. Based on the TelK-DNA crystal structure (Aihara et al, 2007), it is conceivable that TelN's CTD1 domain binds all these repeat motifs to recruit multiple TelN proteins and form a protective nucleo-protein assembly. However, this domain is not essential for protection (Fig. EV4B). Moreover, our in vitro data demonstrate that even at low concentrations, TelN efficiently protects cognate DNA sequences from MR degradation (Fig. 2B), which could potentially be explained by the formation of higher-order structures beyond simple DNA binding. Notably, structural analyses of related telomere resolvases, such as TelK from *Klebsiella oxytoca* phage phiKO2, bound to its minimal target DNA sequence (Aihara et al, 2007), reveal a basic nucleoprotein complex that would putatively readily be overcome by MR. We therefore speculate that TelN contributes to the compaction of hairpin telomeres folding into a larger structure, such as the Telomere (T)-loops observed in mammals (Van Ly et al, 2018; Smith et al, 2020). TelN-mediated DNA looping can be envisaged as an architectural solution to protect hairpin telomeres. It would be an effective mechanism to mediate end protection by sequestering and masking the extreme chromosome termini.

## Evolutionary implications

Our findings provide fundamental insights into bacterial telomere biology and broaden our understanding of DNA end protection in prokaryotic systems. The dispensability of most of the TelN C-terminal domain for protection (Fig. 3), which is highly variable among telomere resolvases, raises the possibility that other bacterial telomere resolvases might share similar protective functions in the more N-terminal domains. The parallels between bacterial and eukaryotic telomere protection suggest that conserved strategies have evolved to solve the end protection problem. TelN-mediated protection of N15 hairpin ends is crucial for phage survival and

propagation and is likely a result of evolutionary adaptation, providing valuable insights into phage-host coevolution. In this regard, MR is acting as a phage defense system with TelN providing counter-defense activity, in analogy to the Gam protein (gamma) of bacteriophage lambda that by protein-DNA mimicry counters the DNA repair and defense system RecBCD (and also inhibits MR/SbcCD)(Kulkarni and Stahl, 1989; Wilkinson et al, 2016). These examples illustrate convergent evolutionary strategies where distantly related phages have independently evolved distinct molecular mechanisms to counteract the same fundamental threat from the bacterial DNA repair nucleases MR and RecBCD.

# Methods

### Reagents and tools table

| Reagent/resource | Reference or source | Identifier or catalog number |
| --- | --- | --- |
| **Experimental models** | | |
| List of *B. subtilis* and | | |
| *E. coli* strains | This paper | Table EV1 |
| **Recombinant DNA** | | |
| List of plasmids | This paper | Table EV2 |
| **Antibodies** | | |
| **Oligonucleotides and other sequence-based reagents** | | |
| Oligonucleotide sequence information | This paper | Table EV3 |
| Prime-IT II Random Primer labeling kit | Agilent Technologies | 300385 |
| **Chemicals, enzymes and other reagents** | | |
| ATP solution (100 mM) | Jena Bioscience | NU-1010 |
| Dithiothreitol (DTT) | PanReac AppliChem | A1101 |
| Sodium dodecyl sulfate (SDS) | Sigma-Aldrich | 71736 |
| D-Biotin | Roth | 3822.4 |
| Chloramphenicol | Sigma-Aldrich | C0378 |
| Spectinomycin dihydrochloride | PanReac AppliChem | A3834,0005 |
| T1 Streptavidin-coated Dynabeads (Dynabeads™ MyOne™ Streptavidin T1) | Thermo Fisher Scientific | 65601 |
| Potassium acetate 99% | abcr | AB118624 |
| Rubidium chloride (RbCl) | Sigma-Aldrich | 83979 |
| Calcium chloride dihydrate ($CaCl_2 \cdot 2H_2O$) | Serva | 39551.01 |
| Manganese (II) chloride tetrahydrate ($MnCl_2 \cdot 4H_2O$) | Sigma-Aldrich | M3634 |
| Magnesium chloride hexahydrate ($MgCl_2 \cdot 6H_2O$) | Sigma-Aldrich | M2670 |
| Potassium chloride (KCl) | Sigma-Aldrich | 60130 |
| Sodium chloride (NaCl) | Sigma-Aldrich | 31434-1KG-R |
| Sodium hydroxide (NaOH) | Sigma-Aldrich | 221465 |

| Reagent/resource | Reference or source | Identifier or catalog number |
|---|---|---|
| Glycerol | Sigma-Aldrich | 49770 |
| Trizma base | Sigma-Aldrich | T1503 |
| 3-(N-morpholino) propanesulfonic acid (MOPS) | Sigma-Aldrich | M1254 |
| Ethylenediaminetetraacetic acid (EDTA) | HuberLab | A4892.0500 |
| Tris(2-carboxyethyl) phosphine (TCEP) | Sigma-Aldrich | 646547 |
| 2-mercaptoethanol | Sigma-Aldrich | M-7154 |
| Imidazole | Merck | 1047160250 |
| HiTrap Q HP column 5 mL | Cytiva | 17115401 |
| HisTrap HP 5 mL | Cytiva | 17524802 |
| Superose 6 Increase 10/300 GL | Cytiva | 29091596 |
| HiLoad Superdex 200 column | Cytiva | 28989335 |
| Novex 4-12% Tris-Glycine Gels | Life Technologies | XP04125BOX |
| D-(+)-Xylose | Biochemica | A2241,0500 |
| L-(+)-Arabinose | Sigma-Aldrich | A3256 |
| Agarose | Sigma-Aldrich | A9539 |
| Acrylamide | Huberlab | A0385.0500 |
| Ethidium bromide (10 mg/mL) | Thermo Fisher Scientific | 15585011 |
| Bovine Serum Albumin (BSA) | Sigma-Aldrich | A9418 |
| Protease Inhibitor Cocktail (PIC) | Sigma-Aldrich | P8849 |
| Isopropyl β-d-1-thiogalactopyranoside (IPTG) | Thermo Fisher Scientific | R0393 |
| BsaI-HFv2 | New England Biolabs | R3733L |
| XhoI | New England Biolabs | R0146S |
| RadiantDy™ 632 500 MOB molecular weight ladder | Eurogentec | 051023 |
| Hybond-N+ nylon membranes | GE Healthcare | RPN203B |
| Software | | |
| AlphaFold3 | Jumper et al, Evans et al, | https://alphafoldserver.com/ |
| ChimeraX V1.4 | Goddard et al, Pettersen et al, | https://www.cgl.ucsf.edu/chimerax |
| GraphPad Prism V10.5.0 | N/A | https://www.graphpad.com/ |
| SnapGene software V8.0.3 | N/A | https://www.snapgene.com/ |
| ImageQuant TL 1D V8.1 | GE Healthcare | http://www.gelifesciences.com/en/us/shop/protein-analysis/molecular-imaging-for-proteins/imaging-software/imagequant-tl-8-1-p-00110 |

| Reagent/resource | Reference or source | Identifier or catalog number |
|---|---|---|
| ImageJ software V1.53e | National Institutes of Health (NIH) | https://imagej.nih.gov/ij |
| Other | | |
| Original data | This paper | https://data.mendeley.com/datasets/8trn4bptpt/1 |

## *B. subtilis* strain construction

All strains constructed in this work are derived from the 1A700 isolate (Table EV1). Natural competence was used to engineer strains at the *yoxD rtp proH*, *amyE, cgeD, sbcC, and sbcD* loci by allelic replacement, as described in (Diebold-Durand et al, 2019). Strains were selected on SMG-agar plates under appropriate antibiotic selection at 37 °C. Genotypes were verified for single-colony isolates by PCR and Sanger sequencing (Microsynth) as required.

## Transformation assay in *B. subtilis*

Transformation assays were performed with naturally competent *B. subtilis* cells. To introduce genomic DNA, we used 60–80 ng of DNA from a strain carrying the appropriate chromosomal antibiotic resistance marker. Plates containing transformant colonies were imaged, and colonies were manually counted.

## Viability assessment by dilution spotting

Bacterial strains were initially cultured on Oxoid Nutrient Agar (ONA) plates supplemented with 0.5% xylose. Single colonies were subsequently inoculated into Luria-Bertani (LB) medium containing 0.5% xylose and cultured overnight at 30 °C with continuous agitation. The overnight cultures were subjected to serial 1:10 dilutions, and bacterial density was estimated by measuring optical density at 600 nm (OD600). Fresh LB medium supplemented with 0.5% xylose was inoculated with the overnight culture to an initial OD600 of 0.001. These cultures were incubated at 37 °C until reaching the exponential phase (OD600 ≈ 0.3). For spotting assays, 200 μL of the exponential phase culture was serially diluted in a 96-well plate to achieve dilutions ranging from $10^{-1}$ to $10^{-7}$. Technical duplicates of 5 μL from each dilution were spotted onto ONA plates containing selective conditions. Colony growth was documented by imaging after 19 h of incubation at 37 °C.

## *E. coli* strain construction

A mini-*Tn7* transposon carrying *araC* and different TelN constructs [TelN WT, TelN(Y425F), TelN(Δ445–631), TelN(Δ541–631), and TelN(Δ583–631)] under the $P_{BAD}$ promoter was integrated into a neutral *E. coli* chromosomal locus downstream of *glmS* by triparental mating, as previously described (Bao et al, 1991). This approach generated multiple *E. coli* strains, each expressing a distinct TelN variant (Table EV1).

### E. coli chemically competent cell preparation and transformation

To prepare chemically competent *E. coli* cells, a sterile flask containing the required volume of LB medium was inoculated with a few colonies from a freshly streaked plate and grown at 37 °C with shaking until an OD600 of 0.4–0.6 was reached. The culture was chilled on ice for 15 min and harvested by centrifugation (5000 rpm, 10 min, 4 °C). Cell pellets were gently resuspended in ice-cold TBF I buffer (30 mM potassium acetate, 100 mM RbCl, 10 mM CaCl$_2$, 50 mM MnCl$_2$*4H$_2$O, 15% glycerol, pH 5.8), using a volume equivalent to 0.4× the original culture volume. After 5 min incubation on ice and subsequent centrifugation, pellets were resuspended in ice-cold TBF II buffer (10 mM MOPS, 10 mM RbCl, 75 mM CaCl$_2$, 15% glycerol, and pH 6.5) at 0.02× the original culture volume. Following 15 min incubation on ice, the suspension was aliquoted (100 μL), flash-frozen in liquid nitrogen, and stored at –70 °C. Where indicated, 0.2% arabinose was added to the culture medium.

For transformation experiments, 1 ng of DNA (either linear plasmids with covalently closed hairpin ends or circular plasmids; Table EV2) was added to 100 μL chemically competent cell aliquots. After heat shock treatment at 42 °C for 1 min, cells were recovered in LB medium for 1 h at 37 °C. Transformed cells were then plated on LB agar supplemented with chloramphenicol and, where indicated, 0.02% arabinose. Plates were incubated overnight at 37 °C. Colony counts were determined using ImageJ software, and all experiments were performed in triplicate.

We performed a paired *t*-test analysis to compare linear versus circular plasmid transformation efficiency and indicated the *p* values wherever significant (below 0.05). Given that our experimental design included multiple control conditions with known expected outcomes to validate assay performance, rather than independent exploratory comparisons, we report uncorrected *p*-values as the primary analysis. The inclusion of multiple controls with predictable outcomes reduces the likelihood of false positive interpretations.

### DNA substrate preparation and fluorescent anisotropy assay

Two DNA substrates were used for fluorescence anisotropy measurements; both derived from an 80-nucleotide palindromic oligonucleotide comprising the TelN target sequence *ntelR* followed by its reverse complement. A fluorescein (6FAM) label was attached to the 3′ end of the oligonucleotide (Table EV3):

5'-aattacggaacatatcagcacacaattgcccattatacgcgcgtataatggg-caattgtgtgctgatatgttccgtaatt-[6FAM]-3'

For the 40 bp specific *ntelR* hairpin substrate, the oligonucleotide was subjected to intramolecular annealing in buffer (100 mM NaCl, 10 mM Tris-HCl, pH 8, and 1 mM EDTA) by heating to 95 °C for 1 min, followed by rapid cooling on ice. The 80 bp specific *ntelRR* duplex DNA was prepared using the same oligonucleotide through intermolecular annealing at 25 °C overnight in the same buffer. Fluorescence anisotropy measurements were performed in a reaction buffer containing 25 mM Tris, pH 7.5, 50 mM KCl, 5 mM MgCl$_2$, and 1 mM MnCl$_2$. A series of TelN protein concentrations, ranging from 0 to 9.6 μM, were incubated in the presence of 50 nM fluorescein-labeled DNA substrates for 30 min at room temperature to attain equilibrium. Measurements were recorded using a Synergy Neo Hybrid Multi-Mode Microplate reader equipped with appropriate fluorescence polarization filters. Assays were conducted in black 96-well flat-bottom plates at 25 °C. Data analysis and curve fitting were performed using GraphPad Prism 10 software using non-linear regression to determine binding parameters.

### Purification of the Mre11-Rad50 protein complex

Mre11-Rad50 was purified as described in (Roisné-Hamelin et al, 2024). The Mre11-Rad50 protein complex was expressed in *E. coli* BL21, using a dual vector system (N-terminal 10His-TwinStrep-3C tagged Mre11 and untagged Rad50). Cells were grown in TB medium at 37 °C until OD600 = 1, cooled to 18 °C, and protein expression was induced with 0.5 mM IPTG for 16 h. Cells were harvested and resuspended in lysis buffer (50 mM Tris, pH 7.5, 300 mM NaCl, 5% glycerol, 25 mM imidazole) containing 100 μL of protease inhibitor cocktail and 5 mM β-mercaptoethanol. Cells were lysed by sonication on ice, using a VS70T probe mounted on a SonoPuls unit (Bandelin), at 40% output power for 13 min with pulsing (1 s on/1 s off). After sonication and ultracentrifugation (40,000 g, 45 min), the clarified lysate underwent a three-step chromatographic purification. First, metal affinity chromatography using a HisTrap HP 5 mL column with imidazole gradient elution (25–500 mM); second, anion exchange chromatography on a HiTrap Q 5 mL column eluted with a NaCl gradient (50–1000 mM); and finally, a size-exclusion chromatography on a Superose 6 Increase 10/300 GL column equilibrated with 20 mM Tris-HCl pH 7.5, 250 mM NaCl, and 1 mM TCEP. The peak fractions containing the intact complex were concentrated to ~2.3 mg/mL, flash-frozen in liquid nitrogen, and stored at –70 °C.

### Purification of TelN constructs

TelN constructs [TelN WT, TelN(Y425F), TelN(Δ541–631), TelN(Δ583–631)] with N-terminal His-TwinStrep-3C tags were expressed in *E. coli* BL21 cells. Cultures were grown in TB medium at 37 °C until OD600 = 1, cooled to 20 °C, and protein expression was induced with 0.5 mM IPTG for 16 h. Cells were harvested by centrifugation and resuspended in lysis buffer (50 mM Tris, pH 7.5, 300 mM NaCl, 5% glycerol, 25 mM imidazole) supplemented with 100 μL of protease inhibitor cocktail and 5 mM β-mercaptoethanol. Following sonication and ultracentrifugation (40,000 × *g*, 45 min), the clarified lysate underwent a four-step purification process. First, proteins were purified by affinity chromatography using a HisTrap HP column with imidazole gradient elution (25–500 mM). Then, for TelN WT, TelN(Y425F), and TelN(Δ583–631), overnight tag cleavage was performed during dialysis using 20 mM Tris, pH 7.5, 100 mM NaCl, and 5 mM β-mercaptoethanol supplemented with 500 μL 3C protease. For TelN Δ541–631, tag cleavage was performed overnight without dialysis to prevent precipitation. Finally, proteins were subjected to anion exchange chromatography using a HiTrap Q column with NaCl gradient elution. Finally, size-exclusion chromatography was performed on a HiLoad Superdex 200 column equilibrated with 10 mM Tris, pH 7.5, 200 mM NaCl, 0.1 mM EDTA, and 1 mM DTT. Purified proteins were concentrated to 7.4 mg/mL TelN WT, 7.0 mg/mL TelN(Y425F), 4.0 mg/mL TelN(Δ583–631), and 4.3 mg/mL TelN(Δ541–631), flash-frozen in liquid nitrogen, and stored at −70 °C.

## DNA resolution assay

About 37 nM of 566 bp *ntelRL* DNA duplex, generated by PCR amplification, was incubated with increasing concentrations of TelN proteins (0–1200 nM) in 20 μL reactions containing nuclease buffer (25 mM Tris-HCl, pH 7.5, 50 mM KCl, 5 mM $MgCl_2$, 1 mM $MnCl_2$, 0.1 mg/mL BSA, and 1 mM DTT). Samples were incubated at 30 °C for 30 min, followed by heat inactivation at 75 °C for 5 min. The resulting DNA species were resolved by electrophoresis on 1.5% agarose gels containing ethidium bromide.

## DNA end protection assay

A 566 bp *ntelRL* sequence (*ntelRL* sequence with additional neighboring sequences from N15 phage; Table EV3) was incubated with TelN, generating two hairpin products of 237 and 329 bp using its cleaving-joining activity. The biotinylated and fluorescently labeled 237 bp hairpin substrate was used for subsequent analyses. DNA immobilization was performed using T1 streptavidin-coated Dynabeads (10 μL, 100 μg per reaction). Beads were first pre-equilibrated using 10 volumes of nuclease buffer (25 mM Tris-HCl, pH 7.5, 50 mM KCl, 5 mM $MgCl_2$, 1 mM $MnCl_2$, 0.1 mg/mL BSA, and 1 mM DTT). About 37 nM of biotinylated hairpin DNA was bound to the beads in a 20 μL reaction at 25 °C for 15 min with constant shaking at 850 rpm. Protection assays were conducted by incubating the immobilized hairpin DNA with Mre11-Rad50 tetramer (125 or 500 nM) and increasing concentrations of TelN variants (0–1200 nM). Reactions were performed in nuclease buffer supplemented with 1 mM ATP at 37 °C for 15 min with shaking. Reactions were stopped by adding 80 μL buffer containing 10 mM EDTA and 20 mM Tris. Protected DNA was eluted by adding 100 μL preheated nuclease buffer supplemented with 25 mM biotin and 0.1% SDS at 70 °C for 15 min. The supernatant containing the eluate was separated from the beads using a magnetic rack. A RadiantDy™ 632 500 MOB molecular weight ladder (Eurogentec, LOT# 051023) was included for size determination. Samples were resolved on DNA-denaturing 5% polyacrylamide gels prepared in 1× alkaline buffer (50 mM NaOH and 1 mM EDTA). Gels were processed under light-protected conditions and subsequently scanned using a Typhoon fluorescence imager (GE Healthcare) with dual-channel acquisition: Cy2 channel for fluorescently labeled DNA substrates and Cy5 channel for ladder detection using auto PMT settings for both channels.

For species specificity experiments, 1 nM of fluorescently labeled hairpin DNA substrate was incubated with either 80 nM *E. coli* MR or 80 nM *S. cerevisiae* MRX complemented with 320 nM pSae2. Both MRX and pSae2 were kindly provided by the Cejka lab. TelN was added at three different concentrations (20, 80, and 320 nM) to assess protection against both complexes. The reactions were performed and analyzed following the same protocol described above.

For multi-substrate protection assays, we used three different fluorescently labeled DNA substrates simultaneously: a 329 bp *ntel*-containing hairpin labeled with TAMRA, a biotinylated 237 bp *ntel*-containing hairpin labeled with fluorescein and immobilized on streptavidin beads, and a 411 bp nonspecific linear DNA fragment (random sequence, non-*ntelRL*, non-hairpin DNA) labeled with ATTO 680. The total DNA concentration was maintained at 30 nM across all experiments, with equimolar amounts of each substrate (10 nM each). Protection assays were conducted by incubating the DNA substrates with 500 nM Mre11-Rad50 tetramer and increasing concentrations of TelN WT (0–640 nM). Reactions were performed in nuclease buffer supplemented with 1 mM ATP at 37 °C for 15 min with shaking, followed by the same stopping and elution procedures described above. For each fluorophore (fluorescein, TAMRA, and ATTO 680), the Typhoon fluorescence imager (GE Healthcare) was set with appropriate excitation and emission filter settings. For overlay images, individual fluorescence channels were pseudo-colored (fluorescein: blue, TAMRA: green, and ATTO 680: red) and combined using Adobe Photoshop.

## Southern blot analysis

Genomic DNA was extracted from *B. subtilis* strains using standard methods and digested with BsaI and XhoI restriction enzymes (New England Biolabs) overnight at 37 °C. Digested DNA was separated on 0.8% agarose gels and transferred to nylon membranes (Hybond-N+, GE Healthcare) using alkaline transfer methods. A $^{32}$P-labeled DNA probe targeting the *ntelRL*-spectinomycin resistance gene region was hybridized overnight at 65 °C. Membranes were washed with 2× SSC/0.1% SDS at room temperature and 0.1× SSC/0.1% SDS at 65 °C, then exposed to a phosphor image screen and subsequently scanned on a Typhoon scanner (GE Healthcare). Fragment sizes were determined using DNA molecular weight standards. The analysis focused on BsaI-XhoI fragments, with expected sizes of 5797 bp for intact constructs and 4180 bp + 1621 bp for TelN-cleaved linearized chromosomes.

# Data availability

Unprocessed image files and numeric data (original raw data) are available at Mendeley Data, https://doi.org/10.17632/8trn4bptpt.1 (https://data.mendeley.com/datasets/8trn4bptpt/1). Any additional information required to reanalyze the data reported in this paper is available from the corresponding author upon request.

The source data of this paper are collected in the following database record: biostudies:S-SCDT-10_1038-S44318-025-00593-z.

# Peer review information

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

## Acknowledgements

We are grateful to Nikolai Ravin for generously sharing N15 phage-related reagents. We thank Petr Cejka and his lab for providing MRX and Sae2 proteins, Michael Taschner and Yan Li for help and advice on recombinant protein purification, Hon-Wing Liu for feedback on figure schematics, and Florian Roisné-Hamelin for stimulating discussions and valuable input on data interpretation. We are grateful to Kerri Kobryn and members of the Gruber lab for helpful feedback during manuscript revision. This work was supported by SNSF project funding (320030-227915) to SG.

## Author contributions

**Maya Houmel**: Data curation; Formal analysis; Validation; Investigation; Visualization; Methodology; Writing—original draft; Writing—review and editing. **Nicolas Pellaton**: Investigation; Methodology. **Anna Anchimiuk**: Investigation; Methodology. **Stephan Gruber**: Conceptualization; Supervision; Funding acquisition; Visualization; Writing—review and editing.

Source data underlying figure panels in this paper may have individual authorship assigned. Where available, figure panel/source data authorship is listed in the following database record: biostudies:S-SCDT-10_1038-S44318-025-00593-z.

## Disclosure and competing interests statement

The authors declare no competing interests.

# Expanded View Figures

**Figure EV1. Chromosome linearization in *B. subtilis*.**

(A) Schematic model for *E. coli* MR sensing and processing of chemically diverse DNA ends. Rad50 comprises an ATP-regulated catalytic head harboring DNA-binding activities, two long, protruding antiparallel coiled coils, and an apical zinc-hook dimerization domain. The DNA-binding and processing head module is formed by the Mre11 nuclease dimer and the Nucleotide-Binding Domains (NBDs) of Rad50. In the resting state, the complex adopts an auto-inhibited conformation where the Rad50 ATPase dimer blocks the Mre11 nuclease active site. Upon DNA binding, the complex transitions to a hypothetical scanning state where it searches for DNA ends along the double helix. Detection of a DNA end triggers an ATP-regulated conformational change, forming the cutting state via a ring-to-rod transition, closing the coiled coils onto a single DNA double helix, repositioning the Mre11 dimer to the side of the complex. MR discriminates against circular DNA, as the presence of two DNA strands within the MR ring prevents the closure of the Rad50 coiled coils. (B) *B. subtilis* transformation efficiency as in Fig. 1B but without normalization. (C) Suppressor mutations in the specR–*ntelRL* locus in *B. subtilis*. To test whether the *B. subtilis* chromosome can be linearized, we co-introduced *telN* and its cognate *ntelRL* site. No clones carrying both intact *telN* and *ntelRL* were recovered in otherwise wild-type cells; however, rare suppressor clones emerged carrying mutations in or near *ntelRL*. Two-point mutants (Mutants 1 and 2) contain substitutions within the *ntelRL* site itself (red boxes). Two additional suppressor clones (Mutants 3 and 4) contain larger deletions (511 and 225 bp, respectively) spanning the specR–*ntelRL* locus. (D) Schematic representation of the chromosomal region analyzed by Southern blotting. The upper panel shows the wild-type (WT) configuration with expected fragments: 4884 bp between BsaI and XhoI (black arrows) and 6755 bp between the two BsaI sites in case of partial XhoI digestion (red dashed arrows). The position of the probe used for hybridization is indicated by a black bar targeting the *ntelRL*-spectinomycin region. The lower panel shows the expected outcome after chromosome linearization. Due to partial XhoI digestion observed in our experiments, additional BsaI-BsaI fragments emerge (red dashed arrows): 7668 bp in the circular context. (E) Southern blot analysis of BsaI/XhoI-digested genomic DNA from different *B. subtilis* strains. While our analysis primarily focuses on BsaI-XhoI-digested fragments, partial XhoI digestion resulted in additional bands (marked with red asterisks) representing BsaI-BsaI fragments. Lane 1: WT strain. Lanes 2, 3, and 4: Strains lacking the *B. subtilis* MR complex (ΔMR) and containing the *ntelRL* site inserted adjacent to the spectinomycin resistance gene (specR). Lane 2 (ΔMR + specR_*ntelRL* + P$_{xyl}$-*telN*) and lane 4 (ΔMR + specR_*ntelRL* + P$_{rpsB}$-*telN*). An additional 3492 bp band (*) appears due to partial XhoI digestion in these linearized chromosomes. Lane 3 (ΔMR + specR_*ntelRL* + P$_{rpsB}$-*gfp*): Strain containing the *ntelRL* insertion shows a 5797 bp BsaI-XhoI band and a 7668 bp band (*) from partial XhoI digestion, demonstrating that chromosome linearization specifically requires TelN expression. (F) Schematic of pLIN_*ntelN* linear plasmid (12.3 kb) with its DNA hairpin ends (*ntelL* and *ntelR*) depicted in the blowout, adapted from (Liu et al, 2022).

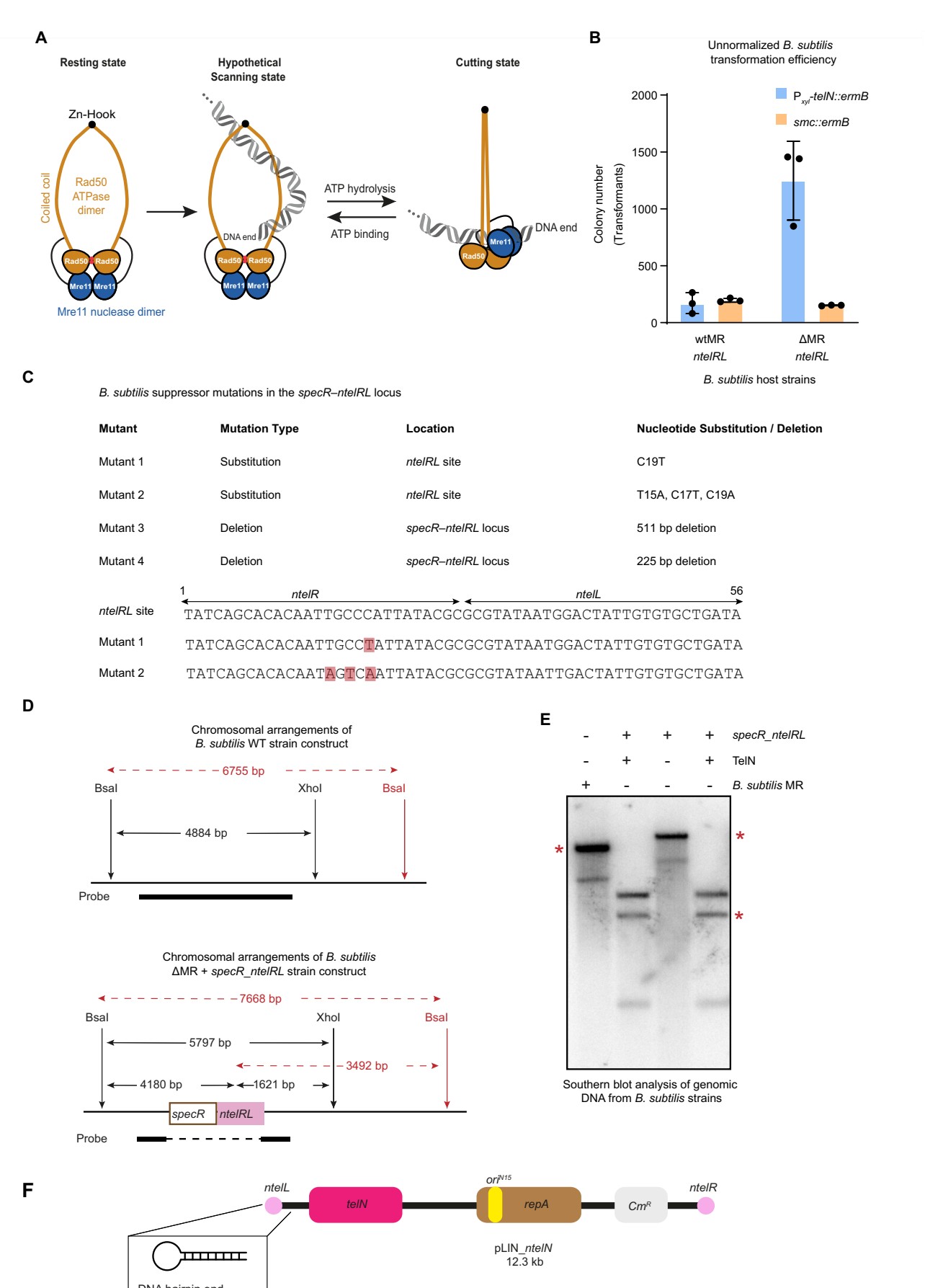

**C** *B. subtilis* suppressor mutations in the *specR–ntelRL* locus

| Mutant | Mutation Type | Location | Nucleotide Substitution / Deletion |
|---|---|---|---|
| Mutant 1 | Substitution | *ntelRL* site | C19T |
| Mutant 2 | Substitution | *ntelRL* site | T15A, C17T, C19A |
| Mutant 3 | Deletion | *specR–ntelRL* locus | 511 bp deletion |
| Mutant 4 | Deletion | *specR–ntelRL* locus | 225 bp deletion |

Southern blot analysis of genomic DNA from *B. subtilis* strains

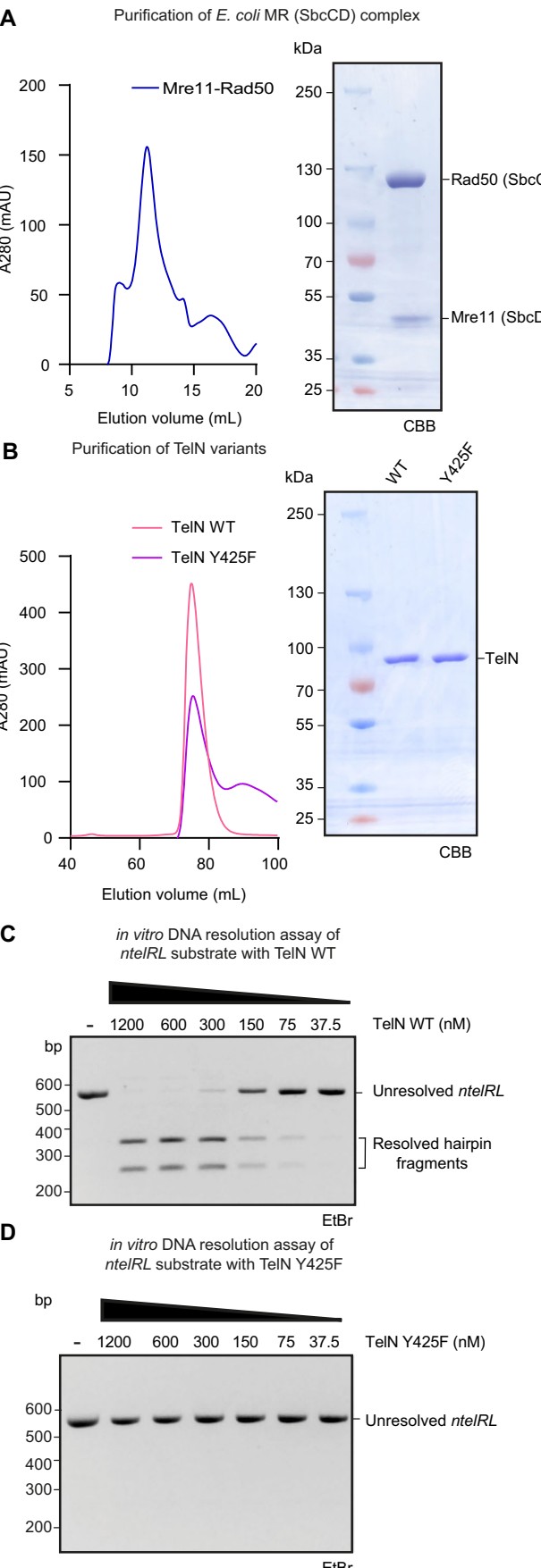

**A** Purification of *E. coli* MR (SbcCD) complex

**B** Purification of TelN variants

**C** *in vitro* DNA resolution assay of *ntelRL* substrate with TelN WT

**D** *in vitro* DNA resolution assay of *ntelRL* substrate with TelN Y425F

**Figure EV2. Protein purification and characterization.**

(**A**, **B**) Left panels: representative size-exclusion chromatography (SEC) elution profile for the purification of (**A**) the *E. coli* MR complex, (**B**) TelN WT and TelN(Y425F) proteins. Right panels: SDS-PAGE profile of purified fractions of (**A**) MR, (**B**) TelN WT, and TelN(Y425F) as visualized by Coomassie brilliant blue (CBB) staining. (**C**, **D**) DNA resolution assays on *ntelRL* DNA substrate (*ntelRL* with additional neighboring sequences from N15 phage) using decreasing concentrations of (**C**) TelN WT and (**D**) TelN(Y425F). DNA species were resolved with a 1.5% ethidium bromide (EtBr) agarose gel.

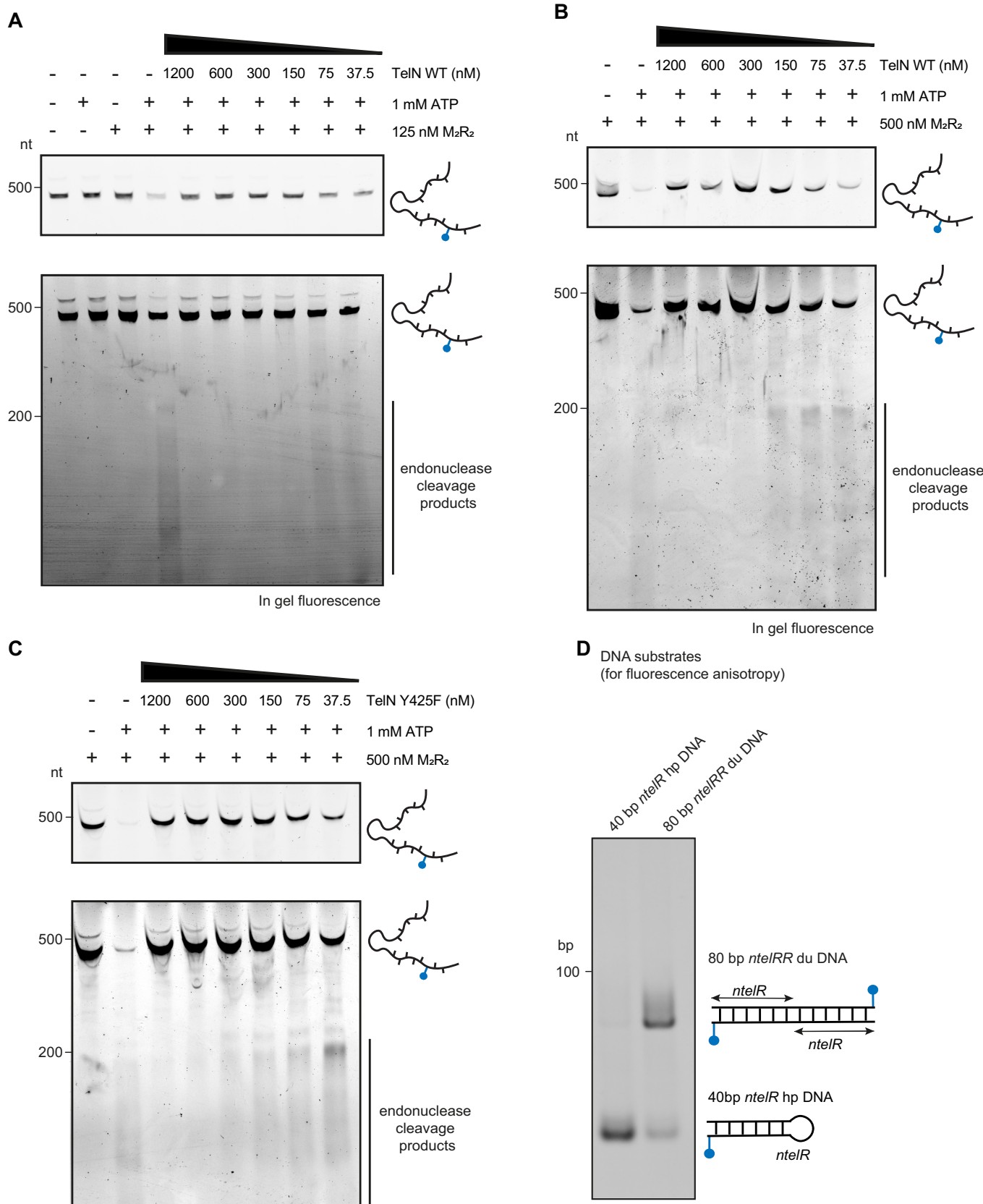

**Figure EV3.  TelN DNA binding and protection assays.**

(A–C) In vitro DNA protection assays using (A) TelN WT with 125 nM $M_2R_2$ (including ATP-only control), (B) TelN WT with 500 nM $M_2R_2$, or (C) TelN(Y425F) with 500 nM $M_2R_2$. About 37 nM of the immobilized *ntelL* hairpin substrate (237 bp) was incubated with the respective $M_2R_2$ concentration and decreasing concentrations of the indicated TelN variant. Products were analyzed by DNA-denaturing PAGE after biotin/SDS elution. Lower panel: representative in-gel fluorescence analysis showing DNA degradation profile. Upper panel: same gel image with exposure optimized to visualize remaining substrate bands. (D) Analysis of fluorescently labeled DNA substrates on a 10% native polyacrylamide gel. The substrates used for binding measurements with TelN WT and TelN(Y425F) include a 40 bp *ntelR* hairpin (hp) DNA and an 80 bp *ntelRR* duplex (du) DNA. Both substrates were derived from the same 80-nucleotide palindromic oligonucleotide, which contains the *ntelR* sequence followed by its reverse complement. The palindromic nature of this sequence allows it to form either a 40 bp hairpin through intramolecular annealing or an 80 bp dsDNA through intermolecular annealing, depending on preparation conditions.

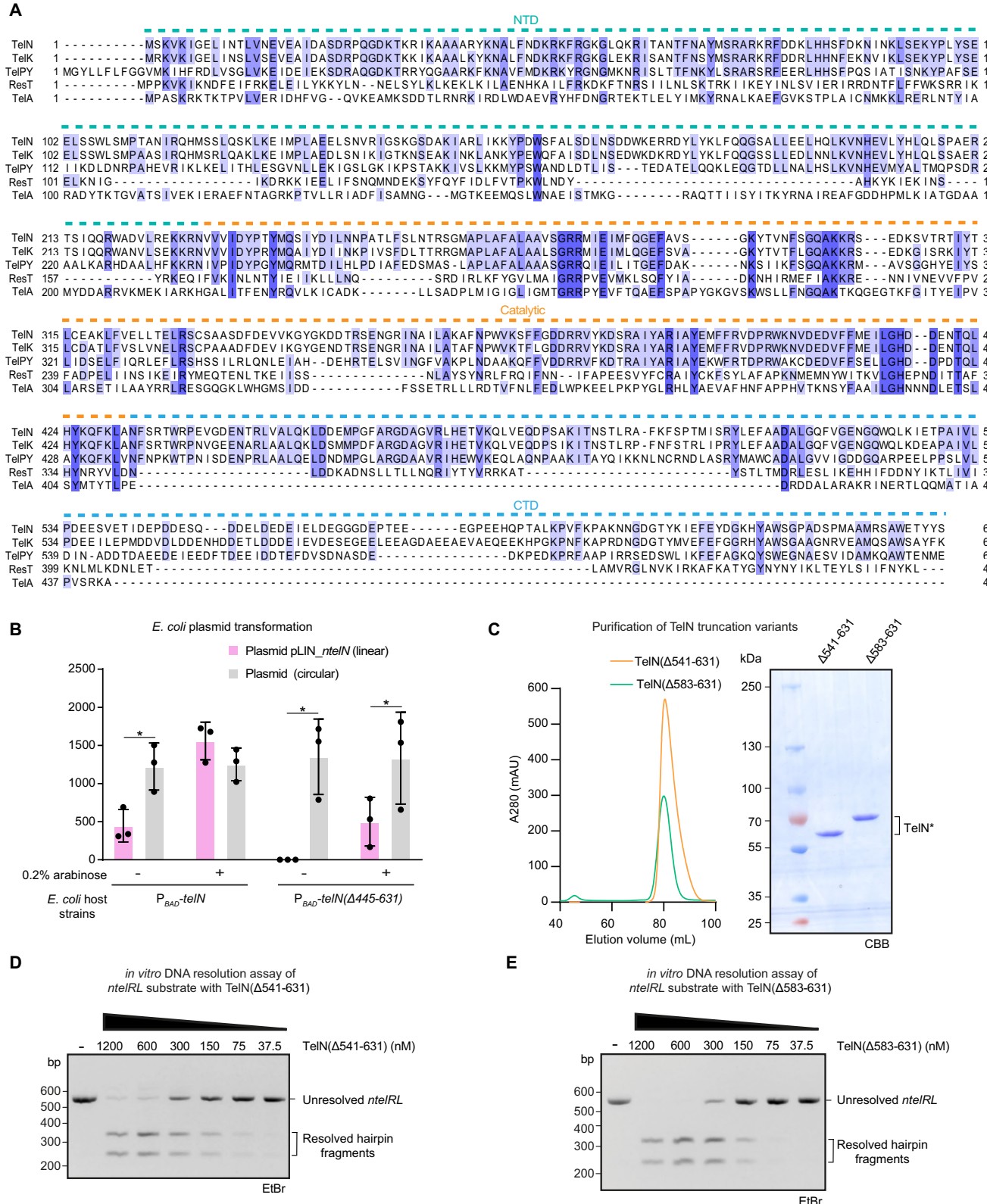

**Figure panels A–E**

A. Sequence alignment of TelN, TelK, TelPY, ResT, TelA showing NTD, Catalytic, and CTD domains.

B. *E. coli* plasmid transformation

C. Purification of TelN truncation variants

D. *in vitro* DNA resolution assay of *ntelRL* substrate with TelN(Δ541-631)

E. *in vitro* DNA resolution assay of *ntelRL* substrate with TelN(Δ583-631)

**Figure EV4.   Design and characterization of TelN truncation mutants.**

(A) Multiple-sequence alignment of telomere resolvases, including TelN from *E. coli* phage N15, TelK from *Klebsiella oxytoca* phage phiKO2, TelPY from *Yersinia* phage PY54, TelA from *Agrobacterium tumefaciens*, and ResT from *Borrelia burgdorferi*. Numbers indicate amino acid positions. The sequence alignment was done on JalView; residues are color-coded based on percentage identity. TelN domains are indicated by colored dashed lines: N-ter domain (NTD, green dashed line), catalytic domain (orange dashed line), and C-ter domain (CTD, blue dashed line). (Uniprot accession numbers: TelN, Q37967; TelK, Q6UAV6; TelPY, Q7Y3Y3; ResT, O50979; TelA, F8KAE9). (B) Counts of chloramphenicol-resistant colonies for the indicated *E. coli* strains after transformation with either pLIN_*ntelN* or a control circular plasmid. $P_{BAD}$-*telN*(Δ445–631): strain with chromosomal integration of arabinose-inducible TelN(Δ445–631). Means and standard deviations from three biological replicates are shown. Asterisks indicate samples with a *p* value, obtained by paired *t*-tests, lower than 0.05. $P_{BAD}$-*telN* without arabinose: $p = 0.0233$; $P_{BAD}$-*telN(Δ445–631)* without arabinose: $p = 0.0420$; $P_{BAD}$-*telN(Δ445–631)* with arabinose: $p = 0.0397$. (C) Left panel: representative size-exclusion chromatography (SEC) elution profiles for the purification of TelN(Δ541–631) and TelN(Δ583–631) truncation proteins. Right panel: SDS-PAGE profiles of purified proteins visualized by Coomassie brilliant blue (CBB) staining. (D, E) DNA resolution assays on *ntelRL* DNA substrate using decreasing concentrations of (D) TelN(Δ541–631) and (E) TelN(Δ583–631). DNA species were resolved on a 1.5% EtBr agarose gel.

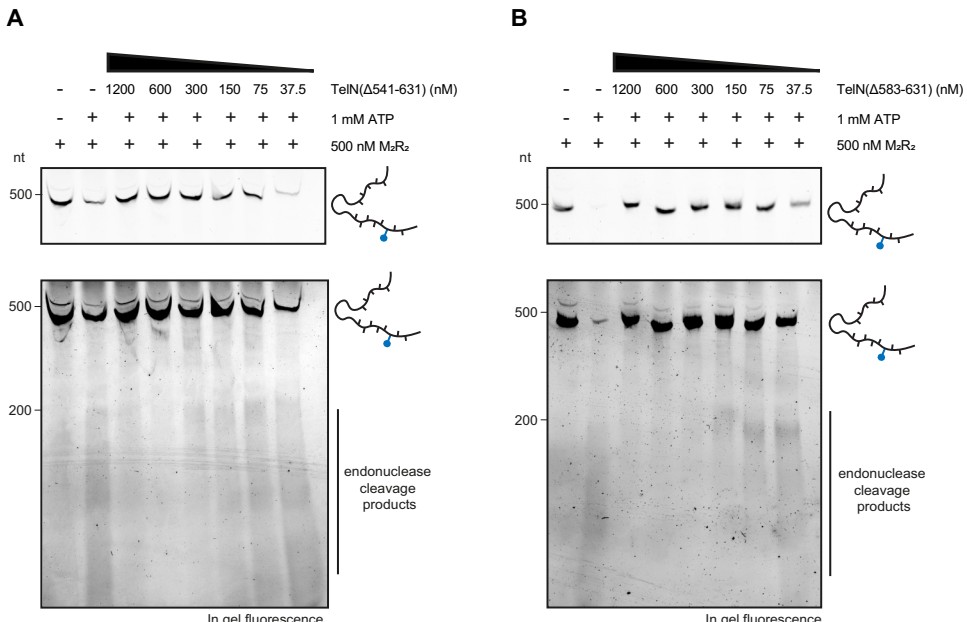

**Figure EV5. In vitro DNA protection assays with Mre11-Rad50.**

(A) TelN(Δ541–631) and (B) TelN(Δ583–631). Products were analyzed by DNA-denaturing PAGE after biotin/SDS elution. Lower panel: representative in-gel fluorescence analysis showing DNA degradation profile. Upper panel: same gel image with exposure optimized to visualize remaining substrate bands.

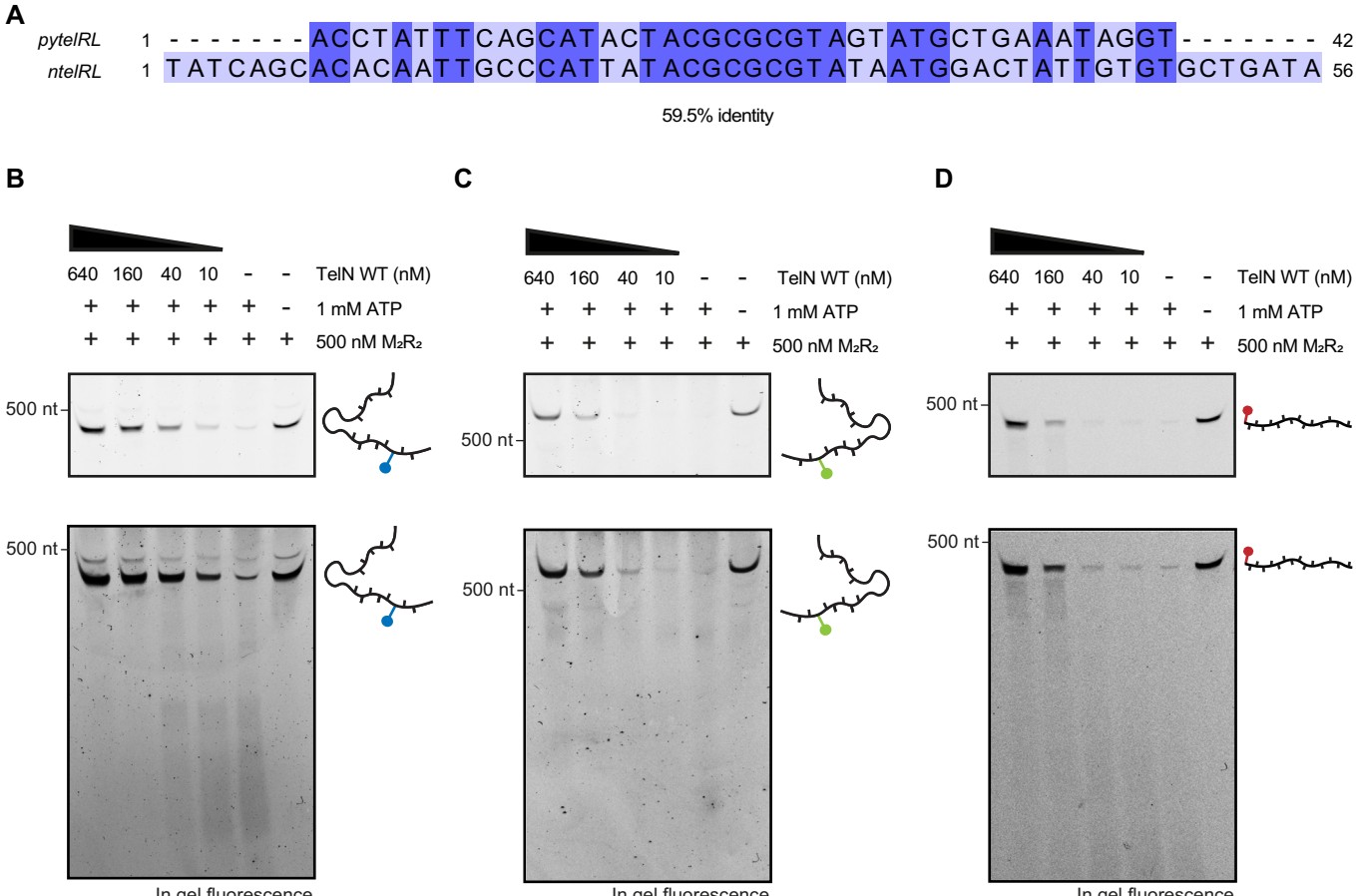

**Figure EV6.  DNA specificity of DNA protection.**

(A) Sequence alignment of the recognition sequences *pytelRL* (42 bp) and *ntelRL* (56 bp). Numbers above the sequences indicate nucleotide positions. The alignment was performed using JalView software; nucleotides are color-coded based on percentage identity. (B, D) Individual gel scans corresponding to the overlay image shown in Fig. 4C for a clearer interpretation of substrate-specific degradation patterns. Lower panels: representative in-gel fluorescence analysis showing DNA degradation profile. Upper panels: same gel image with exposure optimized to visualize remaining substrate bands. The cartoons next to each gel indicate the DNA substrate used in the assay: (B) bead-immobilized 237 bp *ntelL* hairpin DNA substrate labeled with blue fluorescence, (C) 329 bp *ntelR* hairpin DNA substrate labeled with green fluorescence, and (D) 411 bp nonspecific linear DNA substrate labeled with red fluorescence.

