## [Peer Review File · The EMBO Journal]

Phage-Encoded Resolvase TelN Inhibits Bacterial Mre11-Rad50 Nuclease to Protect Hairpin Telomeres

Maya Houmel, Nicolas Pellaton, Anna Anchimiuk, and Stephan Gruber

Corresponding author(s): Stephan Gruber (stephan.gruber@unil.ch)

Review Timeline:

Transfer from Review Commons:	21st Aug 25
Editorial Decision:	17th Sep 25
Revision Received:	24th Sep 25
Accepted:	26th Sep 25

Review
COMMONS

Editor: Hartmut Vodermaier

Transaction Report: This manuscript was transferred to The EMBO JOURNAL following peer review at Review Commons.

Review #1

1. Evidence, reproducibility and clarity:

Evidence, reproducibility and clarity (Required)

This study addresses how the bacterial telomere protein TelN protect telomere ends against the action of the Mre11-Rad50 nuclease (MR). This protection is essential for the stability of hairpin-ended linear plasmid and chromosomes in bacteria but had not been explored before. The authors demonstrates that TelN is necessary and sufficient to block MR-dependent DNA cleavage when bound to its specific telomere sequence. By combining elegant genetics and biochemical approaches, it convincingly shows that TelN-dependent inhibition likely involves a specific interaction between TelN and the MR complex. The manuscript is well written, easy to read and focused on the relevant information. The claims and the conclusions are supported by the data. There is no over-interpretation.

****Comments:****

- Figure 1B, unnormalized transformation efficiency would be useful to show in SI
- Figures 2B, 2C, 3C, 3D, 4C, 5A and 5B: quantification of independent experiments should be added

****Referee cross-commenting****

Perfect for me. It seems that there is a consensus.

2. Significance:

Significance (Required)

This pioneering study provides a very strong basis for a new understanding of telomeres in bacteria and offers fascinating evolutionary perspectives when compared to similar mechanisms active at telomeres in eukaryotic cells.

3. How much time do you estimate the authors will need to complete the suggested revisions:

Estimated time to Complete Revisions (Required)

(Decision Recommendation)

Less than 1 month

4. Review Commons values the work of reviewers and encourages them to get credit for their work. Select 'Yes' below to register your reviewing activity at Web of Science Reviewer Recognition Service (formerly Publons); note that the content of your review will not be visible on Web of Science.

No

Review #2

1. Evidence, reproducibility and clarity:

Evidence, reproducibility and clarity (Required)

The paper is well-presented and well-written throughout. The paper shows convincingly that TelN protects hairpin DNA ends from the activity of SbcCD, presumably providing a protection mechanism for N15 phage DNA in vivo. Furthermore, this protection activity is shown not to require the catalytic (resolvase) activity of TelN, nor its poorly characterised C-terminal domain. The paper also suggests that this inhibition acts both at the level of competition for the DNA hairpin end and at the level of a direct protein:protein interaction between TelN and MR. An (acknowledged) weakness is that there is no real insight into the protein:protein interaction suggested by the experiments shown in Figure 5. Ideally, the protein:protein interaction interface would be identified and mutations in this interface would be shown to reduce hairpin protection.

****Specific comments/questions****

(1) What pathway (in vivo) leads to inactivation of linear hairpin DNA - one suspects that cleavage by SbcCD at the hairpins is probably not the full story. Presumably SbcCD cleavage facilitates further processing by other long range resection systems such as RecBCD, Exo1, RecQ/J etc. Would it be appropriate to view the hairpin as an adaption to protect against these nucleases, which then must be complemented with a mechanism to suppress SbcCD?

(2) Section starting "Direct inhibition of MR by TelN in vitro". What is the word direct supposed to convey here? To me it suggests that the inhibition is via direct interaction of TelN with MR (rather than, for example, a result of competition for the hairpin DNA end)

which is not shown here. Suggest either defining or removing the word direct. This point gains more importance considering that differentiating between inhibition mechanisms becomes a focus of later parts of the paper.

(3) Figure 2B - Why no control lane without MR? - this is a basic control to show that the degradation we are seeing in the absence of TelN is MR-dependent. Formally, as shown, the degradation could be caused by the ATP stock.

(4) Why not use *B. subtilis* SbcCD for the species specificity experiment? Also, is it not surprising that TelN yielded zero protection against MRX given that the DNA sequence specificity experiments above suggest competition for DNA substrate is part of the inhibition mechanism?

(5) If the authors felt it appropriate, I thought there was scope for further discussion/introductory material. There are strong parallels here with mechanisms used by phage to protect themselves from the activities of RecBCD, which include both proteins that protect DNA ends like T4 gene 2, as well as proteins that bind directly to RecBCD to inactivate it like lambda Gam. As such, the work here will appeal as much to those interested in bacterial defence systems / phage:host interactions as it does to those interested in telomere biology. Especially significant is the inhibition of DNA end processing factors by lambda Gam since this protein is reported to interact with both RecBCD and SbcCD (PMID: 2531105).

(6) Just a gripe really: it seems to be 'de rigeur' at the moment to re-name bacterial proteins for their human orthologues, presumably to elevate the perceived importance of the work(?), but it is not a practice I think is terribly helpful as it causes issues when searching literature. Minimally it would be great if the authors could ensure they add SbcCD as a keyword for search purposes.

****Referee cross-commenting****

I have nothing to add. The reviewers comments are all broadly positive and consistent.

2. Significance:

Significance (Required)

This is an excellent paper unveiling a phage encoded "counter-defence" mechanism designed to protect phage DNA from degradation. It will be of special interest to those studying telomere biology of phage:host interactions.

3. How much time do you estimate the authors will need to complete the suggested revisions:

Estimated time to Complete Revisions (Required)

(Decision Recommendation)

Less than 1 month

No

Review #3

1. Evidence, reproducibility and clarity:

Evidence, reproducibility and clarity (Required)

****Summary:****

The authors investigate how the N15 phage protelomerase TelN protects linear chromosomes that terminate in hairpin structures (a sort of telomere). In *E. coli* and *B. subtilis* cells, removal or truncation of telN reduces transformation/survival of linear DNA, whereas complementation with full-length or a catalytically inactive TelN restores viability, consistent with TelN playing a non-enzymatic capping function.

In vitro, TelN binds hairpin substrates with moderate affinity and protects them from the nuclease activity of the Mre11/Rad50 complex. The authors propose that TelN originated as an early, sequence-specific barrier against MR-mediated DNA end processing, establishing

fundamental principles of telomere protection that persist from bacteria to eukaryotes.

****Major comments:****

The manuscript convincingly shows that TelN can functionally block the Mre11-Rad50 (MR) nuclease on a hair-pin DNA end in a sequence specific manner (suggesting a physical interaction), but it doesn't directly demonstrate this. A simple pull-down or equilibrium binding method would be useful in proving a physical interaction.

The MR complex requires ATP hydrolysis for resection of DNA ends. It would be a nice addition to the manuscript if the effect of TelN on Rad50 ATPase activity was tested.

The bar plot on Fig 3B indicates that the experiments are performed in triplicate. The statistical significance of the differences between conditions should be determined. The same general comment could be made regarding the quantification of the polyacrylamide gels - how reproducible are these values?

****Minor comments:****

A better explanation of how the gels were quantified should be provided. Were the products included in the analysis, or was it just the decrease in the substrate band that was measured?

I felt like the Results jump rather abruptly from *B. subtilis* chromosome assays to *E. coli* plasmid experiments. Maybe the addition of a few linking sentences would improve this transition.

A comment on the stoichiometry of TelN and genome ends during phage replication would be useful.

2. Significance:

Significance (Required)

****General assessment:****

***Strengths:** A nice combination of genetics and biochemistry convincingly demonstrates that TelN protects linear chromosomes/replicons from MR-dependent degradation independent of its cleavage-ligase activity. It does this by binding to the hairpin DNA ends

in a sequence specific fashion and the species specificity suggests a direct physical interaction, which likely inhibits the nuclease activity of the MR complex

Limitations: The lack of characterization of the putative physical interaction between TelN and the MR complex is considered a weakness.

Advance: The manuscript fills in a mechanistic gap between protelomerase-mediated telomere formation and maintenance by demonstrating a protective/capping role. This is the first quantitative analysis of DNA-end protection from MR nuclease activity by TelN.

Audience: Readers interested in bacterial chromosome biology, DNA repair, the parallels to eukaryotic shelterin will be interesting to the broader telomere and genome-stability communities.

3. How much time do you estimate the authors will need to complete the suggested revisions:

Estimated time to Complete Revisions (Required)

(Decision Recommendation)

Less than 1 month

Yes

Full Revision

Manuscript number: RC-2025-03082

Corresponding author(s): Stephan Gruber

[Please use this template only if the submitted manuscript should be considered by the affiliate journal as a full revision in response to the points raised by the reviewers.]

*If you wish to submit a preliminary revision with a revision plan, please use our "Revision Plan" template. **It is important to use the appropriate template to clearly inform the editors of your intentions.***

1. General Statements [optional]

This section is optional. Insert here any general statements you wish to make about the goal of the study or about the reviews.

We appreciate the constructive and supportive feedback on our manuscript. All three reviewers acknowledged the significance and novelty of our work on bacterial telomere protection. In response to their suggestions, we have conducted the requested experiments and revised the manuscript accordingly. These changes have enhanced the rigor of our study and clarified our interpretations and explanations.

Moreover, we characterized an additional truncation mutant of TeIN (TeIN Δ 445–631), which lacks the two C-terminal domains. Despite this deletion, the mutant retained protection activity (Supplementary Figure S4B), indicating that the remaining regions of the protein are sufficient to confer efficient protection in this assay.

Finally, we removed three sequence alignments (previously Supplementary Figures S6A and S7), as we recognized that the high degree of sequence divergence could hinder proper alignment and potentially lead to misinterpretation.

This section is mandatory. Please insert a point-by-point reply describing the revisions that were already carried out and included in the transferred manuscript.

Reviewer #1 (Evidence, reproducibility and clarity (Required)):

This study addresses how the bacterial telomere protein TeIN protects telomere ends against the action of the Mre11-Rad50 nuclease (MR). This protection is essential for the stability of hairpin-ended linear plasmid and chromosomes in bacteria but had not been explored before. The authors demonstrate that TeIN is necessary and sufficient to block MR-dependent DNA cleavage

Full Revision

when bound to its specific telomere sequence. By combining elegant genetics and biochemical approaches, it convincingly shows that TelN-dependent inhibition likely involves a specific interaction between TelN and the MR complex. The manuscript is well written, easy to read and focused on the relevant information. The claims and the conclusions are supported by the data. There is no over-interpretation.

Comments:

- Figure 1B, unnormalized transformation efficiency would be useful to show in SI

The unnormalized *B. subtilis* transformation efficiency has now been added as new figure panel S1B.

- Figures 2B, 2C, 3C, 3D, 4C, 5A and 5B: quantification of independent experiments should be added

While these DNA protection experiments show a clearly reproducible pattern of DNA degradation, the exact response to TelN titration varies somewhat between experimental replicates. We initially included the quantification of remaining full-length DNA because the corresponding band is hard to discern in the gel image due to pixel saturation. However, we realize now that this may mislead readers to think that the degradation occurs always with the exact same dosage response.

To avoid this, we have decided to remove the quantification and instead show the relevant part of the gel also at higher contrast to better visualize the loss of full-length DNA due to DNA degradation. In addition, we have included replicate experiments carried out at the same MR concentration (125 nM M_2R_2) or at higher concentration (500 nM M_2R_2) in the supplementary material. These examples demonstrate the general reproducibility of the assay.

****Referee cross-commenting****

Perfect for me. It seems that there is a consensus.

Reviewer #1 (Significance (Required)):

This pioneering study provides a very strong basis for a new understanding of telomeres in bacteria and offers fascinating evolutionary perspectives when compared to similar mechanisms active at telomeres in eukaryotic cells.

Reviewer #2 (Evidence, reproducibility and clarity (Required)):

The paper is well-presented and well-written throughout. The paper shows convincingly that TelN

protects hairpin DNA ends from the activity of SbcCD, presumably providing a protection mechanism for N15 phage DNA *in vivo*. Furthermore, this protection activity is shown not to require the catalytic (resolvase) activity of TelN, nor its poorly characterised C-terminal domain. The paper also suggests that this inhibition acts both at the level of competition for the DNA hairpin end and at the level of a direct protein:protein interaction between TelN and MR. An (acknowledged) weakness is that there is no real insight into the protein:protein interaction suggested by the experiments shown in Figure 5. Ideally, the protein:protein interaction interface would be identified and mutations in this interface would be shown to reduce hairpin protection.

Specific comments/questions

(1) What pathway (*in vivo*) leads to inactivation of linear hairpin DNA - one suspects that cleavage by SbcCD at the hairpins is probably not the full story. Presumably SbcCD cleavage facilitates further processing by other long range resection systems such as RecBCD, Exo1, RecQ/J etc. Would it be appropriate to view the hairpin as an adaptation to protect against these nucleases, which then must be complemented with a mechanism to suppress SbcCD?

The reviewer's suggestion that hairpin ends represent a first layer of adaptation against nucleolytic processing is compelling. Hairpin structures inherently resist many exonucleases due to their covalently closed nature (absence of free 3' or 5' ends) but remain vulnerable to MR processing (Connelly et al, 1998, 1999; Saathoff et al, 2018). This creates a scenario where effective telomere protection requires both the structural barrier provided by the hairpin and an active mechanism to suppress MR activity. We have added this perspective to the relevant paragraph in the discussion.

(2) Section starting "Direct inhibition of MR by TelN *in vitro*". What is the word direct supposed to convey here? To me it suggests that the inhibition is via direct interaction of TelN with MR (rather than, for example, a result of competition for the hairpin DNA end) which is not shown here. Suggest either defining or removing the word direct. This point gains more importance considering that differentiating between inhibition mechanisms becomes a focus of later parts of the paper.

By "direct inhibition," we meant that TelN blocks MR nuclease activity without requiring additional cofactors, as demonstrated in this minimal reaction system containing only TelN, MR complex, DNA substrate, and ATP. To avoid ambiguity, we have reworded the corresponding headline and paragraph.

(3) Figure 2B - Why no control lane without MR? - this is a basic control to show that the degradation we are seeing in the absence of TelN is MR-dependent. Formally, as shown, the degradation could be caused by the ATP stock.

We have now included ATP-only control lanes (without MR complex), which show no substrate degradation, confirming that ATP stocks do not contain contaminating nucleases and that the observed degradation is indeed MR-dependent. These controls are included in the supplementary data (Figure S3A) along with additional replicate experiments. Notably, the dose-dependent protection observed at low TelN concentrations (where MR activity is not fully inhibited) provides additional evidence for the specificity of the MR-TelN interaction system, as non-specific nuclease contamination would result in complete substrate degradation regardless of TelN concentration.

(4) Why not use *B. subtilis* SbcCD for the species specificity experiment? Also, is it not surprising that TelN yielded zero protection against MRX given that the DNA sequence specificity experiments above suggest competition for DNA substrate is part of the inhibition mechanism?

We agree that this would be a great addition. We attempted but were unable to purify active *B. subtilis* SbcCD protein despite multiple attempts. The yeast MRX experiment serves the same purpose of demonstrating species specificity and represents a more evolutionarily distant comparison, which strengthens our conclusions about bacterial-specific inhibition.

(5) If the authors felt it appropriate, I thought there was scope for further discussion/introductory material. There are strong parallels here with mechanisms used by phage to protect themselves from the activities of RecBCD, which include both proteins that protect DNA ends like T4 gene 2, as well as proteins that bind directly to RecBCD to inactivate it like lambda Gam. As such, the work here will appeal as much to those interested in bacterial defence systems / phage:host interactions as it does to those interested in telomere biology. Especially significant is the inhibition of DNA end processing factors by lambda Gam since this protein is reported to interact with both RecBCD and SbcCD (PMID: 2531105).

We agree that there are obvious parallels between lambda Gam and TelN as counter-defence factors. This was likely largely missed in previous work because the telomere resolution activity of TelN masked its function in counter-defence. We have added a statement on this matter at the end of the discussion.

(6) Just a gripe really: it seems to be 'de rigueur' at the moment to re-name bacterial proteins for their human orthologues, presumably to elevate the perceived importance of the work(?), but it is not a practice I think is terribly helpful as it causes issues when searching literature. Minimally it would be great if the authors could ensure they add SbcCD as a keyword for search purposes.

We appreciate the reviewer's concern about nomenclature inconsistencies in the literature. We have chosen MR over SbcCD as a more generic term that covers eukaryotes, archaea and lately also bacteria and will hopefully contribute to a more consistent terminology in the literature across the domains of life in the future. Our choice to use "Mre11-Rad50" (MR) for the *E. coli* SbcCD complex is also consistent with prominent recent publications (Käshammer et al., 2019; Gut et al., 2022), explicitly referring to the *E. coli* system as "Mre11-Rad50" while acknowledging the

Full Revision

bacterial designation. To link to previous literature, we made sure that both "SbcCD" and "Mre11-Rad50" are mentioned in the abstract. And, as suggested, we have now also added "SbcCD" to our keyword list to facilitate comprehensive literature searches.

****Referee cross-commenting****

I have nothing to add. The reviewers' comments are all broadly positive and consistent.

Reviewer #2 (Significance (Required):

This is an excellent paper unveiling a phage encoded "counter-defence" mechanism designed to protect phage DNA from degradation. It will be of special interest to those studying telomere biology of phage:host interactions.

Reviewer #3

The authors investigate how the N15 phage protelomerase TelN protects linear chromosomes that terminate in hairpin structures (a sort of telomere). In *E. coli* and *B. subtilis* cells, removal or truncation of telN reduces transformation/survival of linear DNA, whereas complementation with full-length or a catalytically inactive TelN restores viability, consistent with TelN playing a nonenzymatic capping function.

In vitro, TelN binds hairpin substrates with moderate affinity and protects them from the nuclease activity of the Mre11/Rad50 complex. The authors propose that TelN originated as an early, sequence specific barrier against MR mediated DNA end processing, establishing fundamental principles of telomere protection that persist from bacteria to eukaryotes.

Major comments:

The manuscript convincingly shows that TelN can functionally block the Mre11Rad50 (MR) nuclease on a hairpin DNA end in a sequence specific manner (suggesting a physical interaction), but it doesn't directly demonstrate this. A simple pull-down or equilibrium binding method would be useful in proving a physical interaction.

We agree that this would be a valuable addition to the study. We have made several attempts to detect direct interaction by co-immunoprecipitation. However, without success so far. We do not have sufficient material for equilibrium binding methods (yet).

The MR complex requires ATP hydrolysis for resection of DNA ends. It would be a nice addition to the manuscript if the effect of TelN of Rad50 ATPase activity was tested.

We have tested the effect of TelN on Rad50 ATPase activity and found no significant impact under our experimental conditions, possible in line with the lack of stable interaction.

Full Revision

The bar plot on Fig 3B indicates that the experiments are performed in triplicate. The statistical significance of the differences between conditions should be determined. The same general comment could be made regarding the quantification of the polyacrylamide gels - how reproducible are these values?

We performed paired t-test analysis for the following figures and now indicate the p-values wherever significant (below 0.05): Figures 1D, 1E, 3B, 4B and S4B. We used paired t-tests to generally compare linear vs circular plasmid transformation efficiency for each condition. In Figure 4B, which included two different linear DNA constructs, we compared the two linear DNA constructs directly to each other. [Given that our experimental design included multiple control conditions with known expected outcomes to validate assay performance, rather than many independent exploratory comparisons, we report uncorrected p-values as the primary analysis. The inclusion of multiple controls with predictable outcomes reduces the likelihood of false positive interpretations.]

As stated in response to reviewer 1, while the exact values for the DNA degradation profile vary somewhat between experiments (likely due to variations in band quantification – see also response to comment below), the general trends are robust as for example indicated by similar experiments performed with higher MR concentration (500 nM instead of 125 nM M_2R_2 concentrations for all TelN variants) demonstrating reproducibility across different conditions. For Figure 5, however, we are unable to provide additional repeat experiments due to limitations in reagent availability. Considering the robust effect seen with Ec MR controls and the presence of multiple samples in the dilution series, we are nevertheless confident about the conclusion.

Minor comments:

A better explanation of how the gels were quantified should be provided. Were the products included in the analysis, or was it just the decrease in the substrate band that was measured?

As also stated above, we have removed the band quantification and instead show the bands also at different contrast settings.

In our original approach, gel band quantification was performed using ImageQuant TL software (version 8.2.0, GE Healthcare). For each gel, individual lanes were defined using either fixed-width boundaries (95-103 pixels) or automatic edge detection, depending on the gel quality and band definition. Band volumes were calculated using rolling ball background subtraction (radius 180 pixels) with automatic band detection. Substrate degradation was assessed by measuring the integrated density (volume) of the remaining full-length (or near full-length) substrate bands under different treatment conditions. The band volume values were plotted directly to compare substrate levels across treatment groups.

Full Revision

We now present the data as two gel panels: an exposure showing the full reaction profile, and another exposure focusing on the substrate bands to clearly demonstrate dose-dependent protection. Additional replicate experiments including ATP-only controls (confirming no contamination from ATP stocks) and experiments at 500 nM M_2R_2 concentrations, are provided in the supplementary data. This approach provides more direct visualization of the biological phenomenon with comprehensive control validation.

I felt like the Results jump rather abruptly from *B. subtilis* chromosome assays to *E. coli* plasmid experiments. Maybe the addition of a few linking sentences would improve this transition.

Upon re-reading the manuscript we agree with this assertion and have added further information to provide a smoother transition.

A comment on the stoichiometry of TelN and genome ends during phage replication would be useful.

Our *in vitro* data suggest that effective protection can be achieved at relatively low TelN:DNA ratios in vitro, consistent with the notion of formation of stable, protective nucleoprotein structures. We unfortunately do not currently have information on the copy number of TelN per cell or per hairpin end. It is not easy to obtain reliable values for these numbers. However, we can speculate that multiple TelN proteins are present due to the presence of three copies of a DNA sequence motif (binding to CTD1) in each telomeric DNA, consistent with the formation of stable, protective nucleoprotein structures.

Reviewer #3 (Significance (Required)):

General assessment:

Strengths: A nice combination of genetics and biochemistry convincingly demonstrates that TelN protects linear chromosomes/replicons from MR-dependent degradation independent of its cleavage-ligase activity. It does this by binding to the hairpin DNA ends in a sequence specific fashion and the species specificity suggests a direct physical interaction, which likely inhibits the nuclease activity of the MR complex

Limitations: The lack of characterization of the putative physical interaction between TelN and the MR complex is considered a weakness.

Full Revision

Advance: The manuscript fills in a mechanistic gap between protelomerase-mediated telomere formation and maintenance by demonstrating a protective/capping role. This is the first quantitative analysis of DNA-end protection from MR nuclease activity by TelN.

Audience: Readers interested in bacterial chromosome biology, DNA repair, the parallels to eukaryotic shelterin will be interesting to the broader telomere and genome stability communities.

Prof. Stephan Gruber
University of Lausanne
Department of Fundamental Microbiology
Quartier UNIL-Sorge
Batiment Biophore
Lausanne, Vaud 1015
Switzerland

17th Sep 2025

Re: EMBOJ-2025-122235-T
Phage-Encoded TelN Inhibits Mre11-Rad50 to Protect Hairpin Telomeres

Dear Stephan,

Thank you for submitting your revised Review Commons manuscript for consideration by The EMBO Journal. Given the interest of the topic and the supportive reports on the preprint, we decided to essentially treat the study like a regular revision, and therefore returned it directly to one of the original referees. As you can see from their feedback below, they were fully satisfied with the revisions and responses, so we shall be happy to accept the manuscript for EMBO Journal publication, as soon as the remaining editorial and formatting issues have been addressed:

GENERAL:

- Please download and complete our author checklist (link provided below).
- Please provide suggestions for a short 'blurb' text prefacing and summing up the conceptual aspect of the study in two sentences (max. 250 characters), followed by 3-5 one-sentence 'bullet points' with brief factual statements of key results of the paper; they will form the basis of an editor-written 'Synopsis' accompanying the online version of the article. Please also upload a synopsis image, which can be used as a "visual title" for the synopsis section of your paper. The image should be in PNG or JPG format, and please make sure that it remains in the modest dimensions of (exactly) 550 pixels wide and 300-600 pixels high.
- You shall also receive a separate message from our Source Data curation team, with instructions on how to prepare and upload relevant image and numerical raw data.

TEXT:

- Please adjust the order as well as the headers of the different manuscript sections: Title page with complete author information, Abstract, Keywords, Introduction, Results, Discussion, Methods, Data Availability, Acknowledgements, Disclosure and Competing Interests Statement, References, Main Figure Legends, Tables, Expanded Figure Legends.
- Please reduce the number of keywords on the abstract page to five (ideally choosing broad general terms).
- Please note that Materials and Methods need to be described in the main text using our 'Structured Methods' format (for detail, see <https://www.embopress.org/page/journal/14693178/authorguide#structuredmethods>). The in-text "Methods" section should contain method and protocol descriptions (ideally using a step-by-step protocol format to facilitate adoption of the methodologies across labs), while all key reagents, experimental models, software and relevant equipment - including their sources and relevant identifiers - should be listed in a separately uploaded Reagents and Tools Table, a template for which can be downloaded from the above section of our Author Guidelines.
- As we are switching from a free-text author contribution statement towards a more formal statement based on Contributor Role Taxonomy (CRediT) terms, please remove the present Author Contribution section and instead specify each author's contribution(s) directly in the Author Information page of our submission system during upload of the final manuscript. See <https://casrai.org/credit/> for more information.
- Please rename the Competing Interest section into "Disclosure and Competing Interests Statement", in accordance with our updated Guide to Authors (<https://www.embopress.org/competing-interests>)
- Please carefully go through the reference list and make sure that each reference is complete with citation year, journal name, volume/page/locator numbers - some of these informations are currently missing in several entries. Also, please note our

format for citation of bioRxiv preprints, which should be in the text: "(preprint: NAME1 et al, YEAR)"; and in the reference list: "Author NAME1, Author NAME2, ... (YEAR) article title. bioRxiv doi: XXX"

- In the Data Availability section, please include an URL linking to the referenced database(s) in which data have been deposited, and ensure their open accessibility at this point.

DATA:

- Please upload all Figures (main and "supplemental") as individual, image-only files with sufficient resolution/quality for production.

- Please convert the "supplementary" figures into "Expanded View Figures", renaming them "Figure EV1-6" both in the legends and when referencing them in the text.

- Please also rename the "supplementary tables" into "Expanded View Tables" -callout "Table EV1-3"- and upload them as separate XLSX files, with their respective legends always included in a separate "legends" tab of the workbook. Alternatively, if you prefer to keep them organized in a single spreadsheet file, this should be renamed "Dataset EV1" and referenced as such; again containing a separate "legends" tab with informations on its contents.

- Finally, during routine pre-acceptance checks, our data editors have raised the following queries regarding figures, data, and legends, which I would ask you to address (ideally using the Track Changes option to facilitate our checking):

* Please note that the exact p-values are required in the legends of figures 1D, E; 3B, 4B

* Please note that information related to n is required in the legends of figures 2D

* Please note that the measure of center for the error bars needs to be defined in the legend of figure 2D

Should you need additional guidance/feedback regarding this final adjustments, please do not hesitate to contact us directly. Thank you again for the opportunity to consider this work for The EMBO Journal, and I look forward to receiving your final version!

With kind regards,

Hartmut

- a point-by-point response to the referees' comments, with a detailed description of the changes made (as a word file).

- a word file of the manuscript text.

- individual production quality figure files (one file per figure)

- a complete author checklist, which you can download from our author guidelines

(<https://www.embopress.org/page/journal/14602075/authorguide>).

- Expanded View files (replacing Supplementary Information)

- a Reagents and Tools Table as part of the Methods section, which can be downloaded from our author guidelines

(<https://www.embopress.org/page/journal/14602075/authorguide#structuredmethods>)

Revision to The EMBO Journal should be submitted online within 90 days, unless an extension has been requested and approved by the editor; please click on the link below to submit the revision online before 16th Dec 2025:

Link Not Available

Referee #3:

I'm satisfied with the author's responses to my previous comments. Because of the lack of detectable interaction and no ATPase inhibition, I agree with reviewer's #2 assertion that the use of 'direct' was perhaps too strong. The authors have tempered their assertion in the discussion: "This species-specificity suggests that TelN makes specific contacts with the bacterial MR complex, pointing toward a protection mechanism involving direct protein-protein interactions with the E. coli complex", which is appropriate.

Rev_Com_number: RC-2025-03082

New_manu_number: EMBOJ-2025-122235-T

Corr_author: Gruber

Title: Phage-Encoded TelN Inhibits Mre11-Rad50 to Protect Hairpin Telomeres

We thank the reviewer for carefully checking the revised version of our manuscript.

Referee #3:

I'm satisfied with the author's responses to my previous comments. Because of the lack of detectable interaction and no ATPase inhibition, I agree with reviewer's #2 assertion that the use of 'direct' was perhaps too strong. The authors have tempered their assertion in the discussion: "This species-specificity suggests that TelN makes specific contacts with the bacterial MR complex, pointing toward a protection mechanism involving direct protein-protein interactions with the E. coli complex", which is appropriate.

Prof. Stephan Gruber
University of Lausanne
Department of Fundamental Microbiology
Quartier UNIL-Sorge
Batiment Biophore
Lausanne, Vaud 1015
Switzerland

26th Sep 2025

Re: EMBOJ-2025-122235R
Phage-Encoded Resolvase TelN Inhibits Bacterial Mre11-Rad50 Nuclease to Protect Hairpin Telomeres

Dear Stephan,

Thank you for submitting your final revised manuscript (including the latest checklist and text updates) for our consideration. I am pleased to inform you that we have now accepted it for publication in The EMBO Journal.

I have still added one minor modification to the text - making the title slightly more explicit and accessible to a wider readership (see above, hope this is fine with you!

With kind regards,

Hartmut
